# Fungi, bacteria and oomycota opportunistically isolated from the seagrass, *Zostera marina*

Cassandra L. Ettinger[1,2]*, Jonathan A. Eisen[1,2,3]

1 Genome Center, University of California, Davis, CA, United States of America, 2 Department of Evolution and Ecology, University of California, Davis, CA, United States of America, 3 Department of Medical Microbiology and Immunology, University of California, Davis, CA, United States of America

* clettinger@ucdavis.edu

## Abstract

Fungi in the marine environment are often neglected as a research topic, despite that fungi having critical roles on land as decomposers, pathogens or endophytes. Here we used culture-dependent methods to survey the fungi associated with the seagrass, *Zostera marina*, also obtaining bacteria and oomycete isolates in the process. A total of 108 fungi, 40 bacteria and 2 oomycetes were isolated. These isolates were then taxonomically identified using a combination of molecular and phylogenetic methods. The majority of the fungal isolates were classified as belonging to the classes Eurotiomycetes, Dothideomycetes, and Sordariomycetes. Most fungal isolates were habitat generalists like *Penicillium* sp. and *Cladosporium* sp., but we also cultured a diverse set of rare taxa including possible habitat specialists like *Colletotrichum* sp. which may preferentially associate with *Z. marina* leaf tissue. Although the bulk of bacterial isolates were identified as being from known ubiquitous marine lineages, we also obtained several Actinomycetes isolates and a *Phyllobacterium* sp. We identified two oomycetes, another understudied group of marine microbial eukaryotes, as *Halophytophthora* sp. which may be opportunistic pathogens or saprophytes of *Z. marina*. Overall, this study generates a culture collection of fungi which adds to knowledge of *Z. marina* associated fungi and highlights a need for more investigation into the functional and evolutionary roles of microbial eukaryotes associated with seagrasses.

## Introduction

Despite their global importance in terrestrial systems, the diversity, function, evolution, and global importance of fungi in the marine environment remains understudied. There are only ~1100 currently accepted species of marine fungi despite estimates that true diversity is much higher, at 10,000 or more species [1, 2]. It is well known that fungi play vital roles in land plant health and fitness (e.g. as pathogens or endophytes), and although much less is known about fungi in aquatic ecosystems, it is thought they have important roles in organic matter degradation and food web dynamics [3]. Thus, it is likely that fungi engage in similarly vital functional roles when associated with marine plants, like seagrasses.

**Data Availability Statement:** The consensus sequences were deposited at GenBank for the fungal ITS2-28S rRNA gene regions under accession no. MN543905-MN544012, for the bacterial 16S rRNA gene under accession no.

MN931878-MN931917, and for the oomycete 28S rRNA gene under accession no. MN944508-MN944509. Fungal 28S rRNA gene alignments and phylogenies generated for this manuscript were deposited to Dryad (doi: 10.25338/B8HS5Z). All additional data and bioinformatic code used or generated here are contained within the manuscirpt and/or Supporting Information files.

**Funding:** This work was supported by grants from the UC Davis H. A. Lewin Family Fellowship and the UC Davis Center for Population Biology to CLE. The funders had no role in study design, data collection and analysis, decision to publish, or preparation of the manuscript.

**Competing interests:** JAE is on the Scientific Advisory Board of Zymo Research, Inc. CLE declares that the research was conducted in the absence of any commercial or financial relationships that could be construed as a potential conflict of interest. This does not alter our adherence to PLOS ONE policies on sharing data and materials.

Seagrasses are fully submerged marine angiosperms and are foundation species in coastal ecosystems. Seagrass beds perform important ecosystem services and can store carbon over very long timescales in their above and below ground tissues and in surrounding sediments (i.e. "blue carbon") [4]. Unfortunately, seagrass beds are threatened by human-related activities such as pollution, climate change and coastal development, and restoration efforts thus far have been mostly ineffective [5]. In addition to their global ecological importance, seagrasses also have a unique evolutionary history. Sometimes referred to as the "whales of the plant world", seagrasses are a paraphyletic group of multiple lineages that convergently adapted to the marine environment between 70 and 100 million years ago [6, 7]. There are only ~60 species of seagrass compared to the ~250,000 species of terrestrial flowering plants, a testament to the strict selective pressure posed by re-entry to the marine environment. This work focuses on one widespread seagrass species, *Zostera marina*, which occurs throughout much of the Northern Hemisphere.

Previous work has characterized the composition and structure of the bacterial community associated with *Z. marina* and found the community to be distinct for different seagrass tissues (e.g., roots, leaves, rhizomes) [8–10]. Many of the abundant bacteria found associated with *Z. marina* are thought to have important functions related to nitrogen and sulfur cycling [10–15] and several culture-dependent studies have obtained bacterial isolates associated with *Z. marina*, ranging from ubiquitous marine lineages to putative sulfate-reducers [16–25].

In comparison, less is known about the fungal community associated with *Z. marina* and seagrasses generally. Culture-based studies have found fungi associated with leaves, roots and rhizomes of seagrasses, but there is little agreement between studies about the taxonomic composition of these communities within and between seagrass species [26–42]. Recently culture-independent studies of seagrass-associated fungi have more thoroughly investigated the diversity of these microorganisms and highlighted a need to further understand factors affecting their biogeography and community dynamics [43–46]. However, these studies were severely hampered by a lack of representation of fungal sequences from the marine environment in public databases and found that taxonomic assignments could not be made for many fungal sequences associated with seagrasses. This suggests both a need to expand molecular knowledge of marine and seagrass-associated fungi in public databases and that seagrasses may harbor diverse and understudied fungal lineages.

Fungi and bacteria are not the only microbes associated with *Z. marina* and there are many other understudied microorganisms that likely have important roles in the seagrass ecosystem. For example, one culture-independent effort sought to investigate the composition of the entire eukaryote community associated with *Z. marina*, and found that the bacterial and eukaryotic epibiont communities were highly correlated [9]. Additionally, oomycetes [47, 48], protists [49], and viruses [50] have all been cultured in association with *Z. marina* and seagrass wasting disease is thought to be caused by the heterokont, *Labyrinthula zosterae* [51].

Here we used a culture-dependent survey followed by molecular and phylogenetic identification to (i) obtain and identify a diverse collection of fungi associated with *Z. marina*, (ii) place this fungal collection in the phylogenetic context of isolates obtained from other seagrass surveys, and (iii) compare and contrast the composition of this fungal collection to high throughput sequencing results of the composition of the fungal community associated with *Z. marina* from the same location.

## Methods

### Sample collection and isolation

*Zostera marina* tissues were collected under California Department of Fish and Wildlife Scientific Collecting Permit # SC 4874 granted to Dr. John J. Stachowicz. Individual *Z. marina*

plants and associated sediment were collected opportunistically from Westside Point (GPS: 38˚19'10.67"N, 123˚ 3'13.71"W) in Bodega Bay, CA during several sampling trips (October 2017, May 2018, July 2018, August 2018 and January 2019) at low tide using a 2.375 inch diameter modified PVC pipe as described in Ettinger & Eisen [43]. Generally, 2–3 cores were obtained per sampling trip. Bulk plant tissue from multiple *Zostera marina* plants of varying ages was also collected during these trips using gloves and placed in sterile plastic bags for use as both an inoculation source and for inclusion in media recipes. Seawater was also collected in autoclaved 1 L nalgene bottles for use in media recipes. All samples were kept cold on ice in a dark cooler for transport back to the lab. Plant tissues, sediment and seawater were stored at 4˚C until plating could occur which happened within 4–24 hours of collection.

Plant tissues, sediment and seawater were plated on a variety of different media types. Only "green" leaf tissue was used as an isolation source. Generally for seagrass tissues (leaf, root or rhizome) this involved, (1) rinsing the tissue with autoclaved nanopure water to remove loosely associated sediment for ~30 sec, (2) using flame sterilized scissors to cut ~1 cm pieces of tissue, (3) placing a subset of these tissue segments directly on plates using flame sterilized tweezers (1–3 segments / plate), (4) taking another subset of tissue segments and placing these segments into 1.5 mL centrifuge tubes with 1 mL of autoclaved nanopure water, (5) vortexing the 1.5 mL centrifuge tubes for ~30 sec, (6) either smashing tissue segments using a sterile pestel or leaving the segments intact, and (7) directly plating intact tissue segments on media using flame sterilized tweezers (1–3 segments / plate) and pipetting 350 μL of wash liquid or of smashed tissue directly on plates. A further subset of tissue segments were subjected to a bleach treatment or were surface cleaned following step (2) above. For the bleach treatment, this involved taking tissue segments and, (1) immersing segments for 5 min in 1 mL 0.5% NaOCl (~10% bleach), (2) then in 1 mL of 95% EtOH for 1 min, (3) then in 1 mL autoclaved nanopure water for 3 min, and (4) directly plating intact bleached tissue segments on media using flame sterilized tweezers (1–3 segments / plate). For the surface cleaned tissues, this involved taking tissue segments and, (1) immersing segments in 500 μL 95% ethanol for ~5 sec, (2) then in 500 μL 0.5% NaOCl ($\sim$10% bleach) for 2 min, (3) then in 500 μL 70% ethanol for 2 min, (4) then rinsing segments with autoclaved nanopure water for 1 min, and (5) directly plating intact surface cleaned tissue segments on media using flame sterilized tweezers (1–3 segments / plate). For sediment this process involved, (1) placing sediment into 1.5 mL centrifuge tubes with 1 mL of autoclaved nanopure water, (2) vortexing the tubes for ~30 sec, and (3) then pipetting 350 μL of sediment suspension directly onto plates. For seawater this process involved pipetting 350 μL of seawater directly onto plates.

A variety of media recipes were used to try to obtain a diverse collection of fungal isolates. These media included 1% tryptone agar (10 g tryptone, 10 g agar, 1 L distilled water), potato dextrose agar (PDA), potato carrot agar (PCA), palm oil media (12 g agar, 10 g dextrose, 10 g yeast extract, 3 g peptone, 2 g L-arginine, 10 mL Tween80,10 mL palm oil, 1 L distilled water, final pH: 8.0), lecithin media (12 g agar, 10 g dextrose, 10 g yeast extract, 3 g peptone, 2 g L-arginine, 10 mL Tween80, 0.7 g lecithin, 1 L distilled water, final pH: 8.0), malt extract agar (MEA; 30 g malt extract, 15 g agar, 1 L distilled water, final pH: 5.5), glucose yeast peptone agar (GYPA; 15 g agar, 5 g yeast extract, 5 g peptone, 40 g glucose,1 L distilled water), and a *Zostera marina* agar (20 g of leaves in 100 mL of 0.45 μM Millipore filtered natural aged seawater heated up to 60˚C for 30 min, 18 g agar, 0.45 μM Millipore filtered natural aged seawater make up volume to 1L) inspired by Agar Posidonia from Panno et al. [33]. A variety of salt amendments were used including: adding no salt, adding varying amounts of instant ocean (8 g, 16 g, or 32 g) or substituting distilled water for 0.45 μM Millipore filtered natural aged seawater. All media was amended with 50 mg/mL ampicillin, with some media batches also amended with 50 mg/mL trimethoprim or 50 mg/mL streptomycin. Additionally, some media

batches also included the addition of 5 g/L dehydrated crushed *Z. marina* leaf tissue. For the exact media conditions each isolate was grown on see S1 Table.

Plates were wrapped in parafilm to prevent contamination and incubated at room temperature (e.g. as in [27, 42]) in the dark (e.g. as in [26, 32]) in a cabinet drawer for a minimum of 4 weeks (e.g. as in [26, 27, 32, 33]), up to a maximum of 12 weeks. Plates were observed every 2–3 days for fungal growth. Fungal isolates were then sterilely subcultured onto new plates and the process repeated until we were confident we had a single isolate. We were confident when we had subcultured the organism three times each with consistent morphology and no signs of contamination. During the isolation process, all parent plates and subcultures for an organism were stored at 4˚C for comparative purposes. Plates with contamination were tossed (e.g. with a morphology inconsistent with what had been previously observed or colonies not near tissues or areas that were streaked).

### DNA extraction, Polymerase chain reaction (PCR) and Sanger sequencing

DNA was extracted from isolates using the MoBio PowerSoil DNA Isolation kit (MO BIO Laboratories, Inc., Carlsbad, CA, United States) with minor changes to the manufacturer's protocol as follows. To improve fungal lysis, samples were heated at 70˚C for 10 minutes between steps 4 and 5. For step 5, samples were bead beaten on the homogenize setting for 2 minutes using a mini-bead beater (BioSpec Products). For a subset of isolates DNA was instead extracted with either the Qiagen Plant DNeasy (QIAGEN, Hilden, Germany), the Qiagen DNeasy PowerSoil Pro Kit (QIAGEN, Hildren, Germany) or the Zymo Xpedition Fungal/Bacterial DNA Mini Prep (Zymo Research Inc, Irvine, CA, United States) according to the manufacturer's instructions. The reason for the discrepancy between which DNA extraction kit was used is that we initially tried several different DNA extraction kits, before finding that the MoBio PowerSoil DNA Isolation kit provided the best DNA yield and subsequently, extracting isolates only with that kit moving forward. For the DNA extraction kit used for each isolate see S1 Table.

Polymerase chain reaction (PCR) was performed using Taq DNA Polymerase (QIAGEN, Hilden, Germany). Initially, PCR was performed on DNA from all isolates to amplify the fungal ITS-28S rRNA gene region. For isolates where PCR was not successful after three attempts, we then attempted to amplify the bacterial 16S rRNA gene. A few samples that had successful amplification for the bacterial 16S rRNA gene had close matches in NCBI GenBank to oomycete mitochondria, so in these cases we then attempted to amplify the oomycete 28S rRNA gene.

The fungal ITS-28S rRNA gene region was obtained using the ITS5 [52] and LR3 [53] primer set, the bacterial 16S rRNA gene was obtained using the 27F [54] and 1391R [55] primer set, and the oomycete 28S rRNA gene was obtained using the LR0R [56] and Un-Lo28S1220 [57] primer set. When amplifying the fungal ITS-28S rRNA gene region, PCR was performed with the following conditions: 95˚C for 5 minutes, 35 cycles at 94˚C for 30 seconds, 52˚C for 15 seconds, 72˚C for 1 minute, and a final extension at 72˚C for 8 minutes [58]. When amplifying the bacterial 16S rRNA gene, PCR was performed with the following protocol: 95˚C for 3 minutes, 40 cycles at 95˚C for 15 seconds, 54˚C for 30 seconds, 72˚C for 1 minute and 30 seconds, and a final extension at 72˚C for 5 minutes (modified from [59]). When amplifying the oomycete 28S rRNA gene, PCR was performed with the following protocol: 94˚C for 4 minutes, 35 cycles at 94˚C for 30 seconds, 57˚C for 30 seconds, 72˚C for 30 sec, and a final extension at 72˚C for 10 minutes (adapted from Bourret *et al.* [60]).

PCR products were visualized on 2% agarose E-gels (Invitrogen, Carlsbad, CA, United States). PCR products were then purified using the Nucleospin Gel and PCR kit (QIAGEN, Hilden, Germany) and quantified using the Qubit dsDNA HS Assay Kit (Invitrogen, Carlsbad, CA, United States). The PCR products were sequenced using the Sanger method by the UC

Davis College of Biological Sciences <sup>UC</sup>DNA Sequencing Facility (http://dnaseq.ucdavis.edu/). The resulting ABI files were visualized and consensus sequences were generated using seqtrace v. 0.9.0 [60] following the Swabs to Genomes workflow [59]. Consensus sequences for the PCR products were deposited at NCBI Genbank under the following accession no. MN543905-MN544012 for the fungal ITS-28S rRNA gene region, MN931878-MN931917 for the bacterial 16S rRNA gene, and MN944508-MN944509 for the oomycete 28S rRNA gene.

## Taxonomic analyses

Preliminary taxonomic assignment of sequences from the PCR products generated above were obtained by comparing the best results (or "top match") across three methods to obtain a consensus assignment. The three methods included (1) using NCBI's Standard Nucleotide BLAST's megablast option against the nr/nt database with default settings for all isolates and against the 16S ribosomal RNA sequence database for bacterial isolates, (2) using the Ribosomal Database Project (RDP) classifier with the appropriate respective database (e.g. the 16S rRNA training set for bacteria, the Fungal LSU, WARCUP and UNITE datasets for fungi, the Fungal LSU for oomycetes) and default settings, (3) using the SILVA Alignment, Classification and Tree (ACT) service with the appropriate database (SSU for bacteria, LSU for fungi and oomycetes) and default settings [61–65]. Taxonomic assignments for isolates and associated isolation conditions were then imported into R (v. 3.6.0) for visualization and analysis using the following packages: ggplot2 (v. 3.2.1), dplyr (v. 0.8.4), reshape (v. 0.8.8), patchwork (v. 1.0.0), and tidyverse (v. 1.3.0) [66–70] (S1 File).

## Phylogenetic analyses of fungal isolates

Sequences closely related to the fungal ITS-28S rRNA gene PCR products generated above were identified using NCBI's Standard Nucleotide BLAST's megablast option with default settings to further confirm fungal taxonomy through phylogenetic placement (S2 Table). Additionally, we wanted to place the *Z. marina* associated fungal isolates in the context of the phylogenetic diversity of available other seagrass-associated fungal isolates. To this end, we performed a literature search to obtain, to our knowledge at the time of the search, all available 28S rRNA sequences obtained from seagrass associated fungal isolates for inclusion in phylogenetic analyses (S3 Table) [26, 27, 42, 71, 72]. Finally, to provide a further framework for these phylogenies, as well as appropriate outgroup taxa, we downloaded the available 28S rRNA sequences previously used in James et al. [73, 74] (S4 Table).

Using the sequences listed in Tables 1 and S2–S4, we generated four different sequence alignments, (1) an alignment to investigate seagrass isolates in the Basidiomycota and Mucoromycota, (2) an alignment to investigate seagrass isolates in the Eurotiomycetes class in the Ascomycota phylum, (3) an alignment to investigate seagrass isolates in the Sordariomycetes class in the Ascomycota phylum, and (4) an alignment to investigate seagrass isolates in the Dothideomycetes class in the Ascomycota phylum.

Each of the four sequence alignments was generated using MAFFT (v. 7.402) [75] with default parameters on the CIPRES Science Gateway web server [76]. The alignments were trimmed using trimAl (v.1.2) with the -gappyout method [77] and then manually inspected with JalView [78]. Sequence alignments were then further trimmed to the D1/D2 regions of the 28S rRNA gene with trimAl using the select option (e.g. Basidiomycota / Mucoromycota alignment {614–2899 }, Eurotiomycetes alignment {0–569 }, Sordariomycetes alignment {501–1224 }, and Dothideomycetes alignment {0–429, 993–1755 }). Spurious sequences (e.g. sequences which contained few or no nucleotides after trimming) were then removed with trimAl using -resoverlap .75 -seqoverlap 50. The resulting alignments contained: 80 sequences

**Table 1. Fungi isolated from the seagrass, *Zostera marina*.**

| Strain | Isolation Source | Class | Order | Putative Taxonomy | GenBank Accession (ITS-LSU) | Genus includes known marine fungi | Genus detected in ITS amplicon data |
|---|---|---|---|---|---|---|---|
| CLE116 | Leaf | Dothideomycetes | Capnodiales | *Cladosporium* sp. | MN543969 | yes | yes |
| CLE118 | Leaf | Dothideomycetes | Capnodiales | *Cladosporium* sp. | MN543970 | yes | yes |
| CLE127 | Leaf | Dothideomycetes | Capnodiales | *Cladosporium* sp. | MN543975 | yes | yes |
| CLE152 | Leaf | Dothideomycetes | Capnodiales | *Cladosporium* sp. | MN543985 | yes | yes |
| CLE37 | Leaf | Dothideomycetes | Capnodiales | *Cladosporium* sp. | MN543925 | yes | yes |
| CLE39 | Leaf | Dothideomycetes | Capnodiales | *Cladosporium* sp. | MN543926 | yes | yes |
| CLE109 | Root | Dothideomycetes | Capnodiales | *Cladosporium* sp. | MN543962 | yes | yes |
| CLE14 | Root | Dothideomycetes | Capnodiales | *Cladosporium* sp. | MN543914 | yes | yes |
| CLE90 | Root | Dothideomycetes | Capnodiales | *Cladosporium* sp. | MN543951 | yes | yes |
| CLE157 | Seawater | Dothideomycetes | Capnodiales | *Cladosporium* sp. | MN543992 | yes | yes |
| CLE121 | Sediment | Dothideomycetes | Capnodiales | *Cladosporium* sp. | MN543973 | yes | yes |
| CLE103 | Leaf | Dothideomycetes | Capnodiales | *Ramularia* sp. | MN543956 | no | yes |
| CLE164 | Leaf | Dothideomycetes | Capnodiales | *Ramularia* sp. | MN544001 | no | yes |
| CLE32 | Leaf | Dothideomycetes | Capnodiales | *Ramularia* sp. | MN543922 | no | yes |
| CLE81 | Leaf | Dothideomycetes | Capnodiales | *Ramularia* sp. | MN543944 | no | yes |
| CLE89 | Leaf | Dothideomycetes | Capnodiales | *Ramularia* sp. | MN543950 | no | yes |
| CLE158 | Rhizome | Dothideomycetes | Capnodiales | *Ramularia* sp. | MN543993 | no | yes |
| CLE160 | Rhizome | Dothideomycetes | Capnodiales | *Ramularia* sp. | MN543996 | no | yes |
| CLE1 | Root | Dothideomycetes | Capnodiales | *Ramularia* sp. | MN543907 | no | yes |
| CLE111 | Root | Dothideomycetes | Capnodiales | *Ramularia* sp. | MN543964 | no | yes |
| CLE112 | Root | Dothideomycetes | Capnodiales | *Ramularia* sp. | MN543965 | no | yes |
| CLE122 | Sediment | Dothideomycetes | Capnodiales | *Ramularia* sp. | MN543974 | no | yes |
| CLE104 | Leaf | Dothideomycetes | Dothideales | *Aureobasidium* sp. | MN543957 | yes | yes |
| CLE102 | Leaf | Dothideomycetes | Pleosporales | Pleosporales sp. | MN543955 | NA | NA |
| CLE3 | Leaf | Dothideomycetes | Pleosporales | Pleosporales sp. | MN543909 | NA | NA |
| CLE55 | Leaf | Dothideomycetes | Pleosporales | Pleosporales sp. | MN543927 | NA | NA |
| CLE56 | Leaf | Dothideomycetes | Pleosporales | Pleosporales sp. | MN543942 | NA | NA |
| CLE57 | Leaf | Dothideomycetes | Pleosporales | Pleosporales sp. | MN543928 | NA | NA |
| CLE159 | Root | Dothideomycetes | Pleosporales | Pleosporales sp. | MN543995 | NA | NA |
| CLE2 | Rhizome | Dothideomycetes | Pleosporales | Pleosporales sp. | MN543908 | NA | NA |
| CLE101 | Leaf | Eurotiomycetes | Eurotiales | *Penicillium* sp. | MN543954 | yes | yes |
| CLE12 | Leaf | Eurotiomycetes | Eurotiales | *Penicillium* sp. | MN543912 | yes | yes |
| CLE128 | Leaf | Eurotiomycetes | Eurotiales | *Penicillium* sp. | MN543976 | yes | yes |
| CLE129 | Leaf | Eurotiomycetes | Eurotiales | *Penicillium* sp. | MN543977 | yes | yes |
| CLE13 | Leaf | Eurotiomycetes | Eurotiales | *Penicillium* sp. | MN543913 | yes | yes |
| CLE130 | Leaf | Eurotiomycetes | Eurotiales | *Penicillium* sp. | MN543978 | yes | yes |
| CLE131 | Leaf | Eurotiomycetes | Eurotiales | *Penicillium* sp. | MN543979 | yes | yes |
| CLE133 | Leaf | Eurotiomycetes | Eurotiales | *Penicillium* sp. | MN543981 | yes | yes |
| CLE139 | Leaf | Eurotiomycetes | Eurotiales | *Penicillium* sp. | MN543983 | yes | yes |
| CLE151 | Leaf | Eurotiomycetes | Eurotiales | *Penicillium* sp. | MN543984 | yes | yes |
| CLE163 | Leaf | Eurotiomycetes | Eurotiales | *Penicillium* sp. | MN544000 | yes | yes |
| CLE17 | Leaf | Eurotiomycetes | Eurotiales | *Penicillium* sp. | MN543916 | yes | yes |
| CLE171 | Leaf | Eurotiomycetes | Eurotiales | *Penicillium* sp. | MN544004 | yes | yes |
| CLE172 | Leaf | Eurotiomycetes | Eurotiales | *Penicillium* sp. | MN544005 | yes | yes |
| CLE174 | Leaf | Eurotiomycetes | Eurotiales | *Penicillium* sp. | MN544007 | yes | yes |
| CLE175 | Leaf | Eurotiomycetes | Eurotiales | *Penicillium* sp. | MN544008 | yes | yes |

*(Continued)*

**Table 1.** (Continued)

| Strain | Isolation Source | Class | Order | Putative Taxonomy | GenBank Accession (ITS-LSU) | Genus includes known marine fungi | Genus detected in ITS amplicon data |
|---|---|---|---|---|---|---|---|
| CLE20 | Leaf | Eurotiomycetes | Eurotiales | *Penicillium* sp. | MN543918 | yes | yes |
| CLE34 | Leaf | Eurotiomycetes | Eurotiales | *Penicillium* sp. | MN543923 | yes | yes |
| CLE35 | Leaf | Eurotiomycetes | Eurotiales | *Penicillium* sp. | MN543924 | yes | yes |
| CLE42 | Leaf | Eurotiomycetes | Eurotiales | *Penicillium* sp. | MN544012 | yes | yes |
| CLE62 | Leaf | Eurotiomycetes | Eurotiales | *Penicillium* sp. | MN543933 | yes | yes |
| CLE66 | Leaf | Eurotiomycetes | Eurotiales | *Penicillium* sp. | MN543936 | yes | yes |
| CLE73 | Leaf | Eurotiomycetes | Eurotiales | *Penicillium* sp. | MN543940 | yes | yes |
| CLE83 | Leaf | Eurotiomycetes | Eurotiales | *Penicillium* sp. | MN543946 | yes | yes |
| CLE84 | Leaf | Eurotiomycetes | Eurotiales | *Penicillium* sp. | MN543947 | yes | yes |
| CLE95 | Leaf | Eurotiomycetes | Eurotiales | *Penicillium* sp. | MN543953 | yes | yes |
| CLE132 | Leaf | Eurotiomycetes | Eurotiales | *Penicillium* sp. | MN543980 | yes | yes |
| CLE106 | Rhizome | Eurotiomycetes | Eurotiales | *Penicillium* sp. | MN543959 | yes | yes |
| CLE107 | Rhizome | Eurotiomycetes | Eurotiales | *Penicillium* sp. | MN543960 | yes | yes |
| CLE108 | Rhizome | Eurotiomycetes | Eurotiales | *Penicillium* sp. | MN543961 | yes | yes |
| CLE113 | Rhizome | Eurotiomycetes | Eurotiales | *Penicillium* sp. | MN543966 | yes | yes |
| CLE114 | Rhizome | Eurotiomycetes | Eurotiales | *Penicillium* sp. | MN543967 | yes | yes |
| CLE115 | Rhizome | Eurotiomycetes | Eurotiales | *Penicillium* sp. | MN543968 | yes | yes |
| CLE145 | Rhizome | Eurotiomycetes | Eurotiales | *Penicillium* sp. | MN543994 | yes | yes |
| CLE155 | Rhizome | Eurotiomycetes | Eurotiales | *Penicillium* sp. | MN543988 | yes | yes |
| CLE25 | Rhizome | Eurotiomycetes | Eurotiales | *Penicillium* sp. | MN543919 | yes | yes |
| CLE26 | Rhizome | Eurotiomycetes | Eurotiales | *Penicillium* sp. | MN543920 | yes | yes |
| CLE41 | Rhizome | Eurotiomycetes | Eurotiales | *Penicillium* sp. | MN544011 | yes | yes |
| CLE85 | Rhizome | Eurotiomycetes | Eurotiales | *Penicillium* sp. | MN543948 | yes | yes |
| CLE110 | Root | Eurotiomycetes | Eurotiales | *Penicillium* sp. | MN543963 | yes | yes |
| CLE15 | Root | Eurotiomycetes | Eurotiales | *Penicillium* sp. | MN543915 | yes | yes |
| CLE161 | Root | Eurotiomycetes | Eurotiales | *Penicillium* sp. | MN543997 | yes | yes |
| CLE162 | Root | Eurotiomycetes | Eurotiales | *Penicillium* sp. | MN543998 | yes | yes |
| CLE58 | Root | Eurotiomycetes | Eurotiales | *Penicillium* sp. | MN543929 | yes | yes |
| CLE59 | Root | Eurotiomycetes | Eurotiales | *Penicillium* sp. | MN543930 | yes | yes |
| CLE60 | Root | Eurotiomycetes | Eurotiales | *Penicillium* sp. | MN543931 | yes | yes |
| CLE68 | Root | Eurotiomycetes | Eurotiales | *Penicillium* sp. | MN543937 | yes | yes |
| CLE119 | Sediment | Eurotiomycetes | Eurotiales | *Penicillium* sp. | MN543971 | yes | yes |
| CLE120 | Sediment | Eurotiomycetes | Eurotiales | *Penicillium* sp. | MN543972 | yes | yes |
| CLE138 | Sediment | Eurotiomycetes | Eurotiales | *Penicillium* sp. | MN543982 | yes | yes |
| CLE167 | Sediment | Eurotiomycetes | Eurotiales | *Penicillium* sp. | MN544003 | yes | yes |
| CLE173 | Sediment | Eurotiomycetes | Eurotiales | *Penicillium* sp. | MN544006 | yes | yes |
| CLE18 | Sediment | Eurotiomycetes | Eurotiales | *Penicillium* sp. | MN543917 | yes | yes |
| CLE64 | Sediment | Eurotiomycetes | Eurotiales | *Penicillium* sp. | MN543935 | yes | yes |
| CLE69 | Sediment | Eurotiomycetes | Eurotiales | *Penicillium* sp. | MN543938 | yes | yes |
| CLE70 | Sediment | Eurotiomycetes | Eurotiales | *Penicillium* sp. | MN543939 | yes | yes |
| CLE77 | Sediment | Eurotiomycetes | Eurotiales | *Penicillium* sp. | MN543941 | yes | yes |
| CLE80 | Sediment | Eurotiomycetes | Eurotiales | *Penicillium* sp. | MN543943 | yes | yes |
| CLE31 | Seawater | Eurotiomycetes | Eurotiales | *Penicillium* sp. | MN543921 | yes | yes |
| CLE144 | Seawater | Eurotiomycetes | Eurotiales | *Talaromyces* sp. | MN543991 | yes | yes |
| CLE82 | Seawater | Eurotiomycetes | Eurotiales | *Talaromyces* sp. | MN543945 | yes | yes |
| CLE92 | Seawater | Eurotiomycetes | Eurotiales | *Talaromyces* sp. | MN543952 | yes | yes |

*(Continued)*

**Table 1.** (Continued)

| Strain | Isolation Source | Class | Order | Putative Taxonomy | GenBank Accession (ITS-LSU) | Genus includes known marine fungi | Genus detected in ITS amplicon data |
|---|---|---|---|---|---|---|---|
| CLE154 | Rhizome | Microbotryomycetes | Sporidiobolales | *Rhodotorula* sp. | MN543987 | yes | yes |
| CLE88 | Leaf | Sordariomycetes | Glomerellales | *Colletotrichum* sp. | MN543949 | no | yes |
| CLE143 | Rhizome | Sordariomycetes | Glomerellales | *Colletotrichum* sp. | MN543989 | no | yes |
| CLE4 | Rhizome | Sordariomycetes | Glomerellales | *Colletotrichum* sp. | MN543905 | no | yes |
| CLE5 | Rhizome | Sordariomycetes | Glomerellales | *Colletotrichum* sp. | MN543906 | no | yes |
| CLE7 | Leaf | Sordariomycetes | Hypocreales | *Acrostalagmus* sp. | MN543911 | yes | yes |
| CLE63 | Rhizome | Sordariomycetes | Hypocreales | *Emericellopsis* sp. | MN543934 | yes | no |
| CLE105 | Leaf | Sordariomycetes | Hypocreales | Hypocreales sp. | MN543958 | NA | NA |
| CLE153 | Leaf | Sordariomycetes | Hypocreales | Hypocreales sp. | MN543986 | NA | NA |
| CLE61 | Root | Sordariomycetes | Hypocreales | Hypocreales sp. | MN543932 | NA | NA |
| CLE6 | Rhizome | Sordariomycetes | Hypocreales | *Sarocladium* sp. | MN543910 | yes | yes |
| CLE146 | Leaf | Sordariomycetes | Hypocreales | *Trichoderma* sp. | MN543999 | yes | yes |
| CLE165 | Leaf | Tremellomycetes | Filobasidiales | *Naganishia* sp. | MN544002 | yes | yes |
| CLE156 | Rhizome | Ustilaginomycetes | Ustilaginales | *Pseudozyma* sp. | MN543990 | yes | no |
| CLE40 | Rhizome | Ustilaginomycetes | Ustilaginales | *Pseudozyma* sp. | MN544010 | yes | no |
| CLE24 | Leaf | Mucoromycetes | Mucorales | *Absidia cylindrospora* | MN544009 | no | no |

Here we report the taxonomic information for each fungal isolate (Class, Order) and the putative taxonomy, provide the GenBank accession number for the ITS-28S rRNA gene sequence for each isolate, and report on the isolation source the isolate was obtained from (e.g. leaf, rhizome, root, seawater or sediment). We also report on whether the genus of each isolate includes marine fungal representatives based on the consensus compiled in Jones et al. [2] and whether the genus of each isolate was detected in the ITS amplicon data in Ettinger & Eisen [43]. Organisms for which a taxonomic identification below the order level was not possible, have a "NA" value for these columns. It is important to note that there is likely significant biological variation within the genera reported here (e.g. among *Penicillium*), such that finding members of these genera should not be interpreted as meaning that the specific variants isolated here have the same biology as variants found to be member of the same genera in other datasets.

with 614 positions (Basidiomycota / Mucoromycota), 91 sequences with 509 positions (Eurotiomycetes), 96 sequences with 501 positions (Sordariomycetes), and 107 sequences with 563 positions (Dothideomycetes).

JModelTest2 (v. 2.1.10) was run with the number of substitution schemes (-s) set to 3 (JC/F81, K80/HKY, SYM/GTR) and then otherwise default parameters on the CIPRES Science Gateway web server to select a best-fit model of nucleotide substitution for use in phylogenetic analyses for each alignment [79, 80]. The best-fit model based on the Akaike Information Criterion values for all alignments was the GTR + I + G evolutionary model.

Using the CIPRES Science Gateway web server, Bayesian phylogenetic inference for each alignment was performed using MrBayes (v. 3.2.2) with four incrementally heated simultaneous Monte Carlo Markov Chains (MCMC) run over 10,000,000 generations. The analysis stopped early if the optimal number of generations to reach a stop value of 0.01 or less for the convergence diagnostic was achieved [81]. This occurred for the Eurotiomycetes, Sordariomycetes and Dothideomycetes alignments at 2,150,000 generations, 1,375,000 generations and 2,140,000 generations, respectively. The Basidiomycota / Mucoromycota alignment ran for the full 10,000,000 generations, only achieving an average standard deviation of split frequencies of 0.049. The first 25% of trees generated for each alignment were discarded as burn-in and for the remaining trees, a majority rule consensus tree was generated and used to calculate the Bayesian Posterior Probabilities. The resulting phylogenies were then visualized with the ggtree (v. 2.0.1), treeio(v. 1.11.2), ggplot2 (v. 3.2.1), and tidyverse (v. 1.3.0) packages in R (v.

3.6.0) and clade labels were added in Adobe Photoshop CS6 [66, 67, 82–85] (S1 File). Alignments and phylogenies generated here were deposited to Dryad [86].

### Comparisons to ITS amplicon data from Ettinger & Eisen [43]

To compare to high throughput sequencing data associated with *Z. marina* from the same location (Westside Point, Bodega Bay, CA), we utilized an amplicon sequence variant (ASV) dataset previously analysed in Ettinger & Eisen [43]. Specifically, we are using the subset ASV table that was used to investigate differences between bulk sample types. Briefly, this ASV table was previously subset to a depth of 10,000 sequences and included 49 samples from four sample types: leaf epiphytes ($n = 13$), root epiphytes ($n = 14$), rhizome epiphytes ($n = 7$), and sediment ($n = 15$). We then used this ASV to make comparisons to the fungal taxa isolated in this study. To investigate whether fungal genera isolated in this study were also detected in the high throughput sequencing data, we generated a list of the unique genera found in the ASV table and compared it to the list of fungal genera isolated here. To investigate whether the fungal genera isolated in this study were detected from the same *Z. marina* tissues, we collapsed the ASV table to the genus level using the tax_glom function in phyloseq. We then subsampled the ASV table to only include the genera of fungi isolated in this study, transformed this ASV table represent presence / absence and visualized a comparison of the distribution of these genera across sample types (leaf, root, rhizome, sediment) to the distribution of these genera across isolation sources (leaf, root, rhizome, sediment). To investigate the mean relative abundance of the fungal orders isolated in this study in the high throughput sequencing data, we collapsed the ASV table to the order level using the tax_glom function in phyloseq. We then subsampled the ASV table to only include the orders of fungi isolated in this study and visualized the distribution of these orders across sample types (leaf, root, rhizome, sediment). These analyses were performed in R (v. 3.6.0) using the ggplot2 (v. 3.2.1), dplyr (v. 0.8.4), reshape (v. 0.8.8), patchwork (v. 1.0.0), phyloseq (v. 1.30.0) and tidyverse (v. 1.3.0) packages [66–69, 82, 87, 88] (S1 File).

## Results

### Isolation efficacy

A total of 160 plates were initially inoculated, 81 with leaves ($n_{whole} = 44$, $n_{crushed} = 22$, $n_{washes} = 11$, $n_{bleached} = 2$, $n_{surface\ cleaned} = 2$), 38 with rhizomes ($n_{whole} = 13$, $n_{crushed} = 8$, $n_{washes} = 2$, $n_{bleached} = 2$, $n_{surface\ cleaned} = 2$), 27 with roots ($n_{whole} = 13$, $n_{crushed} = 8$, $n_{washes} = 2$, $n_{bleached} = 2$, $n_{surface\ cleaned} = 2$), 4 with seawater, and 10 with sediment (S1 and S2 Figs). Microbial growth was observed on 135 plates (84.4% of all inoculated plates). We subcultured 1–5 organisms from all plates with observed microbial growth. However, we only obtained isolates that met our criteria for putatively being single organisms (e.g. which had been subcultured three times each with consistent morphology and no signs of contamination) from 86 of these plates (63.7% of plates with observed growth, 53.8% of all inoculated plates). No isolates were ultimately obtained from bleached or surface cleaned tissues and only one isolate was obtained from *Zostera marina* agar.

In total 185 putatively anexic microbial isolates were obtained. Of these 185 isolates, we were able to generate PCR products for 176 isolates to send for Sanger sequencing for taxonomic identification. Despite multiple attempts we were unable to generate PCR products for 9 isolates across all primer sets tried here (possibly due to primer mismatch and/or too low concentrations of DNA). Of the 170 isolates where PCR products were sent for sequencing, we received good quality sequencing results for and were able to taxonomically identify 150 isolates. For the 26 isolates where sequencing either failed, was low quality or appeared mixed, 17 appeared to be bacterial in origin, 4 appeared to be fungal and 5 were too poor quality to

identify (e.g. comprised of only N's) based on searches using NCBI's Standard Nucleotide BLAST's megablast option with default settings.

## Taxonomic diversity of fungi isolated from *Z. marina*

In an attempt to cultivate a wide diversity of fungal isolates, we used a variety of media types including several which had been used previously to isolate fungi from seagrasses (e.g. PDA [26, 27, 42], GPYA [33], MEA [32]). A total of 108 fungal isolates were obtained, with the majority cultured from *Z. marina* leaf tissue (n = 51), resulting in a range of morphological diversity (Fig 1). The rest of isolates were cultured from rhizome tissue (n = 23), root tissue (n = 16), associated sediment (n = 13), and seawater (n = 5) (Figs 2 and S3 and S4).

Almost all of the fungal isolates were taxonomically classified as belonging to the Ascomycota (n = 103), with the remaining five isolates classified as Basidiomycota (n = 4) and Mucoromycota (n = 1), respectively (Table 1). Within the Ascomycota, isolates were further identified as being in three classes: the Eurotiomycetes (n = 62), Dothideomycetes (n = 30), and Sordariomycetes (n = 11).

Eurotiomycetes isolates were further taxonomically classified as *Penicillium* sp. (n = 59) and *Talaromyces* sp. (n = 3). Sordariomycetes isolates were putatively classified as *Colletotrichum* sp. (n = 4), *Acrostalagmus* sp. (n = 1), *Emericellopsis* sp. (n = 1), *Sarocladium* sp. (n = 1), *Trichoderma* sp. (n = 1), and unidentified Hypocreales sp. (n = 3). Dothideomycetes isolates were classified as *Cladosporium* sp. (n = 11), *Ramularia* sp. (n = 11), *Aureobasidium* sp. (n = 1), and unidentified Pleosporales sp. (n = 7). Basidiomycota isolates were putatively classified as *Pseudozyma* sp. (n = 2), *Rhodotorula* sp. (n = 1), and *Naganishia* sp. (n = 1). The single Mucoromycota isolate was putatively classified as *Absidia cylindrospora*.

We observed a positive relationship between the number of tissue types and number of media types a fungal genus was isolated from ($R^2 = 0.86$; S5 Fig) which we hypothesize may indicate that some fungal genera are habitat generalists. A similar positive relationship is also observed between the number of tissue types and the number of salt sources ($R^2 = 0.92$) as well as between the number of media types and number of salt sources ($R^2 = 0.87$). However, we did not perform any experiments to confirm this pattern. We also did not always attempt to control for effort (e.g. plating the same number of tissue segments on all media types [S1 and S2 Figs]). Therefore, we suggest that these positive relationships be interpreted with caution.

## Taxonomic diversity of bacteria and oomycota isolated from *Z. marina*

Our intent here was to isolate fungi which was why we included broad spectrum antibiotics in our culturing media to help eliminate bacteria which might be associated with *Z. marina*. However, we still cultivated and identified 40 bacteria and 2 oomycetes. The bacteria are likely naturally resistant to the antibiotics used and the oomycetes, as eukaryotes, are unlikely to be affected by their presence in the media. As with the fungal cultivation results, the majority of bacterial isolates were obtained from *Z. marina* leaf tissue (n = 17). The rest of the bacterial isolates were cultured from rhizome tissue (n = 9), root tissue (n = 7), associated sediment (n = 5), and seawater (n = 2) (S6 Fig).

Bacterial isolates were taxonomically identified as belonging to the Actinobacteria (n = 4), Firmicutes (n = 2), Bacteroidetes (n = 2), and Proteobacteria (n = 33) (Table 2). The two Firmicute isolates were further classified as *Bacillus* sp., the two Bacteroidetes isolates as *Joostella* sp., and the Actinobacteria isolates as *Streptomyces* sp. (n = 2), *Rhodococcus* sp. (n = 1), and *Isoptericola* sp. (n = 1). The Proteobacteria isolates were classified as *Vibrio* sp. (n = 18), *Pseudoalteromonas* sp. (n = 8), *Hafnia* sp. (n = 2), *Pseudomonas* sp. (n = 1), *Shewanella* sp. (n = 1), *Marinomonas* sp. (n = 1), and *Phyllobacterium* sp. (n = 1).

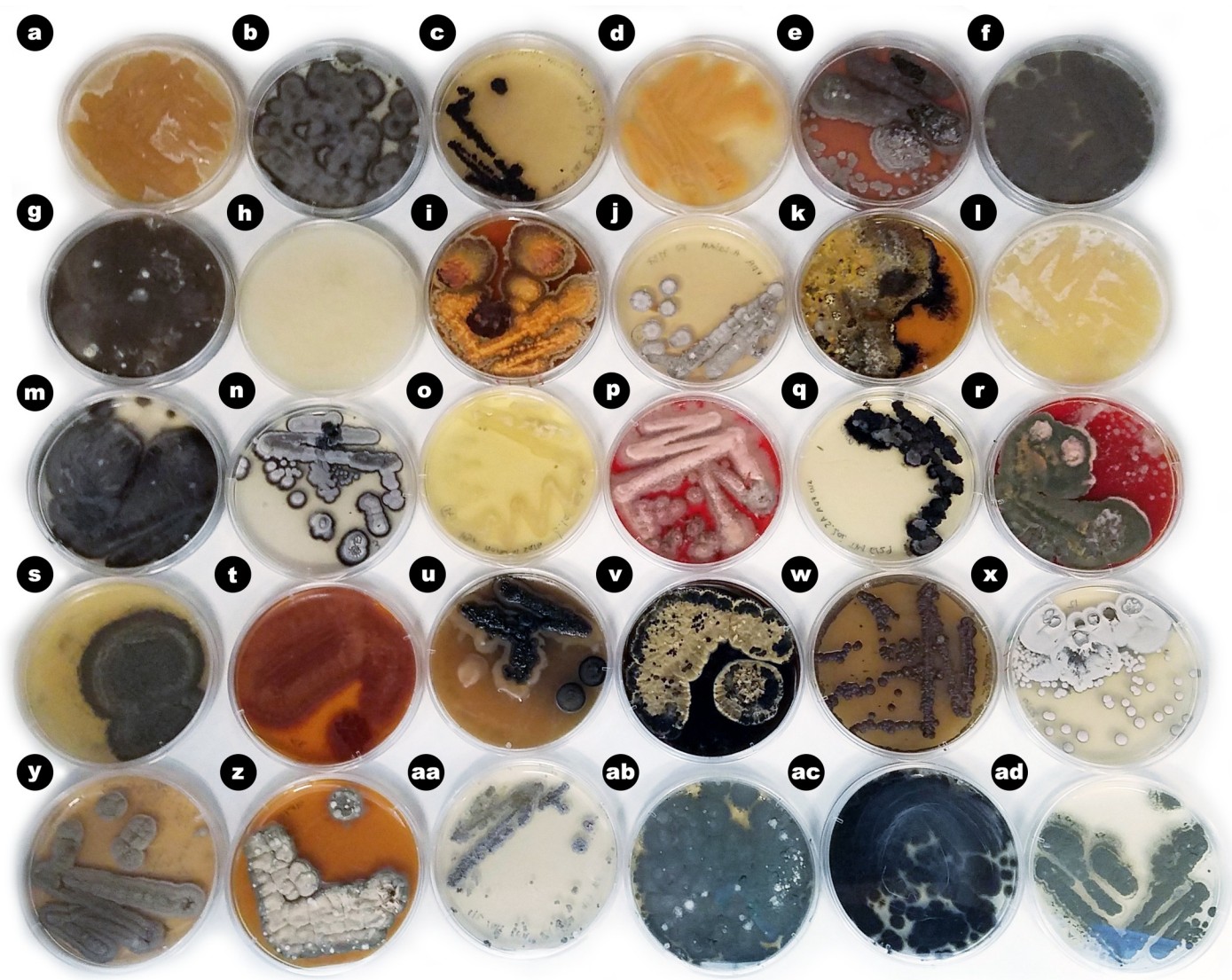

**Fig 1. Microbes isolated from the seagrass, *Zostera marina*.** An example of the morphological diversity of microbial isolates (bacteria, fungi and oomycota) associated with the seagrass, *Z. marina*. Depicted plates were arbitrarily chosen to depict the morphological diversity of the isolates cultured in this study. Putative taxonomy of isolates shown: (a) *Penicillium* sp. CLE73, (b) *Cladosporium* sp. CLE116, (c) *Colletotrichum* sp. CLE5, (d) Hypocreales sp. CLE105, (e) unidentified microorganism, (f) *Penicillium* sp. CLE130, (g) *Penicillium* sp. CLE68, (h) *Halophytophthora* sp. CLE94, (i) *Pleosporales* sp CLE57, (j) unidentified microorganism, (k) *Pleosporales* sp. CLE102, (l) *Penicillium* sp. CLE26, (m) *Cladosporium* sp. CLE118, (n) *Ramularia* sp. CLE122, (o) *Pseudoalteromonas* sp. CLE126, (p) *Talaromyces* sp. CLE92, (q) *Colletotrichum* sp. CLE4, (r) *Talaromyces* sp. CLE82, (s) unidentified microorganism, (t) *Acrostalagmus* sp. CLE7, (u) *Ramularia* sp. CLE1, (v) *Pleosporales* sp. CLE56, (w) *Penicillium* sp. CLE77, (x) *Ramularia* sp. CLE112, (y) *Penicillium* sp. CLE106, (z) unidentified microorganism, (aa) *Streptomyces* sp. CLE117, (ab) *Penicillium* sp. CLE114, (ac) *Cladosporium* sp. CLE127, and (ad) *Penicillium* sp. CLE110. Unidentified microorganisms were unable to be identified using molecular methods (i.e. a PCR product was not successfully generated).

The two oomycete isolates were obtained from *Z. marina* leaf tissue and were both taxonomically identified as *Halophytophthora* sp. (Table 3).

## Phylogenetic comparison of fungal isolates across seagrass species

To confirm fungal isolate identity and investigate if *Z. marina* fungal isolates were closely related to fungal isolates obtained from other seagrass species, we built four phylogenetic trees,

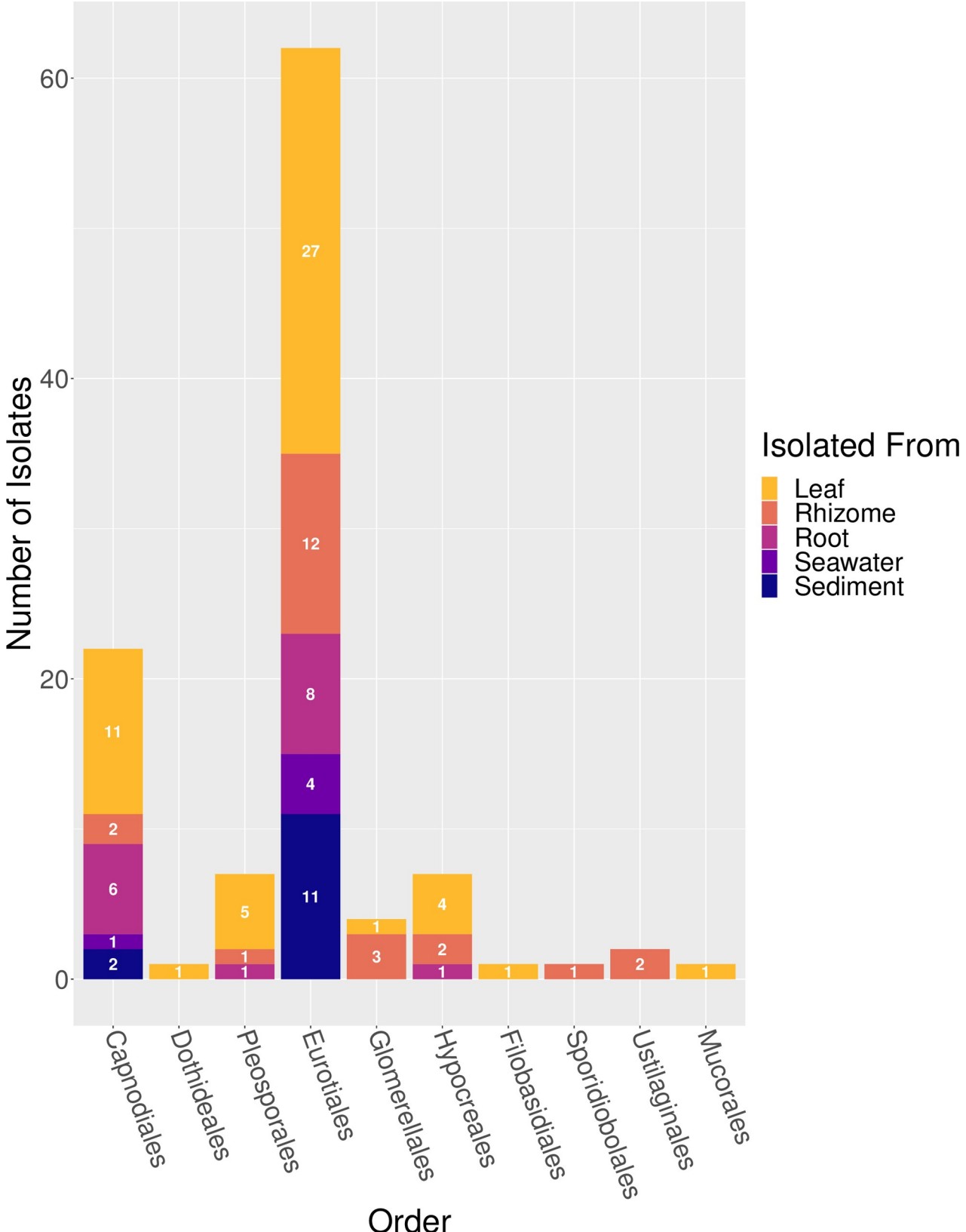

**Fig 2. Distribution of counts of fungal isolates across isolation sources.** A histogram representing the number of fungal isolates grouped by order and colored by isolation source (leaf, rhizome, root, seawater or sediment). The numbers included on each bar represent the count of isolates obtained from that particular isolation source.

1) a phylogeny of seagrass isolates in the Basidiomycota and Mucoromycota (Figs 3 and S7), (2) a phylogeny of seagrass isolates in the Eurotiomycetes class in the Ascomycota phylum (Figs 4 and S8), (3) a phylogeny of seagrass isolates in the Sordariomycetes class in the Ascomycota phylum (Figs 5 and S9), and (4) a phylogeny of seagrass isolates in the Dothideomycetes class in the Ascomycota phylum (Figs 6 and S10). The placements of isolates in these phylogenies were consistent with the taxonomic identities previously determined. Additionally, isolates that were only identified at the order level taxonomically (unidentified Pleosporales sp. and Hypocreales sp.) were not able to be confidently further identified via the phylogenetic methods used here. The closest phylogenetic relatives of unidentified Hypocreales sp. are other unidentified fungi in the order Hypocreales (Fig 5). While the unidentified Pleosporales sp. form an unresolved clade with members in the family Didymellaceae in the order Pleosporales (Fig 6).

We expected to see more shared taxonomic groups and phylogenetic clustering between the fungal isolates of *Z. marina* and those of other seagrass species than was observed in Figs 3–6. Many of the fungal isolates from *Z. marina* did not cluster with fungal isolates that had been previously cultivated in association with other seagrass species. One interpretation might be that each seagrass species harbors a unique fungal community. However, this could also be the result of slight differences in collection protocols, media recipes or other methodology involved in isolating fungi in the compared studies.

The fungal taxa that did have close relatives that were associated with other species included *Penicillium* sp. (Fig 4), *Trichoderma* sp. (Fig 5), *Cladosporium* sp. and *Ramularia* sp. (Fig 6). We note that *Penicillium* sp., *Cladosporium* sp. and *Ramularia* sp. are drivers of the positive relationship observed previously between the number of tissue types and number of media types a fungal genus was isolated from (S5 Fig).

## Comparisons to ITS amplicon sequencing data from Ettinger & Eisen

We compared the diversity of the fungi isolated here to high throughput sequencing data associated with *Z. marina* from the same location (as previously analyzed in Ettinger & Eisen [43]. We found that the fungal genera isolated in this study were generally also detected in the sequencing data (Table 1). Only three genera were not detected in the sequencing data, *Pseudozyma* sp., *Emericellopsis* sp. and *Absidia cylindrospora*. The absence of these genera in the sequencing data may be methodological (e.g. they do not amplify with the primer set used to generate the sequencing data) or biological (e.g. due to seasonal variation).

We then investigated whether the fungal genera isolated in this study were also detected in association with the same types of samples (e.g. leaf, root, rhizome, sediment) in the sequencing data. We observed that many of rare (e.g. not as frequently isolated) genera were not consistently detected on the same sample type with both methods, whereas many of the abundant (e.g. more frequently isolated) genera were detected with both methods (Fig 7). When we looked at the mean relative abundance of the fungal orders isolated in this study in the high throughput sequencing data, we observed that the genera detected using both methods generally were in orders that had higher mean relative abundances in the seagrass ecosystem (S11 Fig). We also observed that some orders such as the Eurotiales (e.g. *Penicillium* sp.) and Capnodiales (e.g. *Cladosporium* sp. and *Ramularia* sp.) had similar mean relative abundances

**Table 2. Bacteria isolated from the seagrass, *Zostera marina*.**

| Strain | Isolation Source | Class | Order | Putative Taxonomy | GenBank Accession (SSU) | Top BLAST match | BLAST % Identity | Top BLAST accession no. |
|---|---|---|---|---|---|---|---|---|
| CLE44 | Leaf | Actinobacteria | Actinomycetales | *Isoptericola* sp. | MN931913 | *Isoptericola halotolerans* | 99.2 | NR_043198.1 |
| CLE150 | Sediment | Actinobacteria | Actinomycetales | *Rhodococcus* sp. | MN931907 | *Rhodococcus erythropolis* | 99.92 | NR_037024.1 |
| CLE117 | Leaf | Actinobacteria | Streptomycetales | *Streptomyces* sp. | MN931916 | *Streptomyces argenteolus* | 99.36 | NR_112300.1 |
| CLE43 | Leaf | Actinobacteria | Streptomycetales | *Streptomyces* sp. | MN931912 | *Streptomyces beijiangensis* | 98.35 | NR_112607.1 |
| CLE16 | Root | Alphaproteobacteria | Rhizobiales | *Phyllobacterium* sp. | MN931909 | *Phyllobacterium loti* | 95.73 | NR_133818.1 |
| CLE136 | Sediment | Bacilli | Lactobacillales | *Bacillus* sp. | MN931897 | *Bacillus mycoides* | 99.61 | NR_036880.1 |
| CLE53 | Sediment | Bacilli | Lactobacillales | *Bacillus* sp. | MN931915 | *Bacillus thuringiensis* | 99.92 | NR_043403.1 |
| CLE8 | Rhizome | Flavobacteria | Flavobacteriales | *Joostella* sp. | MN931878 | *Joostella marina* | 99.3 | NR_044346.1 |
| CLE10 | Root | Flavobacteria | Flavobacteriales | *Joostella* sp. | MN931908 | *Joostella marina* | 99.22 | NR_044346.1 |
| CLE126 | Leaf | Gammaproteobacteria | Alteromonadales | *Pseudoalteromonas* sp. | MN931894 | *Pseudoalteromonas spiralis* | 99.46 | NR_114801.1 |
| CLE71 | Leaf | Gammaproteobacteria | Alteromonadales | *Pseudoalteromonas* sp. | MN931884 | *Pseudoalteromonas spiralis* | 99.52 | NR_114801.1 |
| CLE74 | Leaf | Gammaproteobacteria | Alteromonadales | *Pseudoalteromonas* sp. | MN931886 | *Pseudoalteromonas hodoensis* | 98.42 | NR_126232.1 |
| CLE140 | Rhizome | Gammaproteobacteria | Alteromonadales | *Pseudoalteromonas* sp. | MN931899 | *Pseudoalteromonas hodoensis* | 98.98 | NR_126232.1 |
| CLE141 | Rhizome | Gammaproteobacteria | Alteromonadales | *Pseudoalteromonas* sp. | MN931900 | *Pseudoalteromonas spiralis* | 99.14 | NR_114801.1 |
| CLE142 | Rhizome | Gammaproteobacteria | Alteromonadales | *Pseudoalteromonas* sp. | MN931901 | *Pseudoalteromonas spiralis* | 99.92 | NR_114801.1 |
| CLE98 | Rhizome | Gammaproteobacteria | Alteromonadales | *Pseudoalteromonas* sp. | MN931892 | *Pseudoalteromonas spiralis* | 99.3 | NR_114801.1 |
| CLE147 | Sediment | Gammaproteobacteria | Alteromonadales | *Pseudoalteromonas* sp. | MN931903 | *Pseudoalteromonas hodoensis* | 98.6 | NR_126232.1 |
| CLE47 | Rhizome | Gammaproteobacteria | Alteromonadales | *Shewanella* sp. | MN931882 | *Shewanella surugensis* | 97.78 | NR_040950.1 |
| CLE149 | Seawater | Gammaproteobacteria | Enterobacteriales | *Hafnia* sp. | MN931906 | *Hafnia alvei* | 99.54 | NR_112985.1 |
| CLE87 | Seawater | Gammaproteobacteria | Enterobacteriales | *Hafnia* sp. | MN931890 | *Hafnia alvei* | 99 | NR_112985.1 |
| CLE19 | Leaf | Gammaproteobacteria | Oceanospirillales | *Marinomonas* sp. | MN931910 | *Marinomonas rhizomae* | 97.5 | NR_116233.1 |
| CLE28 | Rhizome | Gammaproteobacteria | Pseudomonadales | *Pseudomonas* sp. | MN931911 | *Pseudomonas sabulinigri* | 97.69 | NR_044415.1 |
| CLE123 | Leaf | Gammaproteobacteria | Vibrionales | *Vibrio* sp. | MN931893 | *Vibrio ostreicida* | 99.15 | NR_133887.1 |
| CLE125 | Leaf | Gammaproteobacteria | Vibrionales | *Vibrio* sp. | MN931917 | *Vibrio lentus* | 99.68 | NR_114982.1 |
| CLE148 | Leaf | Gammaproteobacteria | Vibrionales | *Vibrio* sp. | MN931904 | *Vibrio kanaloae* | 98.53 | NR_114804.1 |
| CLE170 | Leaf | Gammaproteobacteria | Vibrionales | *Vibrio* sp. | MN931902 | *Vibrio alginolyticus* | 99.61 | NR_122050.1 |
| CLE176 | Leaf | Gammaproteobacteria | Vibrionales | *Vibrio* sp. | MN931905 | *Vibrio penaeicida* | 96 | NR_042121.1 |
| CLE29 | Leaf | Gammaproteobacteria | Vibrionales | *Vibrio* sp. | MN931879 | *Vibrio kanaloae* | 99.12 | NR_114804.1 |
| CLE30 | Leaf | Gammaproteobacteria | Vibrionales | *Vibrio* sp. | MN931880 | *Vibrio kanaloae* | 98.65 | NR_114804.1 |
| CLE72 | Leaf | Gammaproteobacteria | Vibrionales | *Vibrio* sp. | MN931885 | *Vibrio alginolyticus* | 98.69 | NR_122050.1 |
| CLE78 | Leaf | Gammaproteobacteria | Vibrionales | *Vibrio* sp. | MN931889 | *Vibrio tasmaniensis* | 99.43 | NR_036929.1 |
| CLE65 | Leaf | Gammaproteobacteria | Vibrionales | *Vibrio* sp. | MN931883 | *Vibrio penaeicida* | 96.66 | NR_042121.1 |
| CLE75 | Rhizome | Gammaproteobacteria | Vibrionales | *Vibrio* sp. | MN931887 | *Vibrio kanaloae* | 99.45 | NR_114804.1 |
| CLE76 | Rhizome | Gammaproteobacteria | Vibrionales | *Vibrio* sp. | MN931888 | *Vibrio kanaloae* | 99.14 | NR_114804.1 |
| CLE134 | Root | Gammaproteobacteria | Vibrionales | *Vibrio* sp. | MN931895 | *Vibrio penaeicida* | 95.89 | NR_042121.1 |
| CLE135 | Root | Gammaproteobacteria | Vibrionales | *Vibrio* sp. | MN931896 | *Vibrio penaeicida* | 95.89 | NR_042121.1 |

(*Continued*)

**Table 2.** (Continued)

| Strain | Isolation Source | Class | Order | Putative Taxonomy | GenBank Accession (SSU) | Top BLAST match | BLAST % Identity | Top BLAST accession no. |
|--------|-----------------|-------|-------|-------------------|------------------------|-----------------|------------------|-------------------------|
| CLE36 | Root | Gammaproteobacteria | Vibrionales | *Vibrio* sp. | MN931881 | *Vibrio tasmaniensis* | 99.68 | NR_036929.1 |
| CLE48 | Root | Gammaproteobacteria | Vibrionales | *Vibrio* sp. | MN931914 | *Vibrio kanaloae* | 98.96 | NR_114804.1 |
| CLE52 | Root | Gammaproteobacteria | Vibrionales | *Vibrio* sp. | MN931891 | *Vibrio penaeicida* | 96.92 | NR_042121.1 |
| CLE137 | Sediment | Gammaproteobacteria | Vibrionales | *Vibrio* sp. | MN931898 | *Vibrio anguillarum* | 98.92 | NR_042509.1 |

Here we report the taxonomic information for each bacterial isolate (Class, Order) and the putative taxonomy, provide the GenBank accession number for the 16S rRNA gene sequence for each isolate, and report on the isolation source the isolate was obtained from (e.g. leaf, rhizome, root, seawater or sediment). We also report the taxonomic identity of the top BLAST match against NCBI's targeted loci 16S ribosomal RNA sequence database, the BLAST % identity to the bacterial isolate and the GenBank accession number for the 16S rRNA gene sequence for the top BLAST match.

across all sample types. While other orders such as Glomerellales (e.g. *Colletotrichum* sp.) had a higher mean relative abundance on one sample type (e.g. leaves).

## Discussion

Here, we generated a fungal collection of 108 isolates expanding understanding of the diversity of *Z. marina* associated fungi, while also underscoring how little we know about these under-studied microorganisms. Generally, the taxonomic diversity observed in our cultivation efforts is consistent with that of other culture-dependent studies which found Eurotiomycetes, Dothideomycetes, and Sordariomycetes to be the main classes of fungi associated with seagrasses [26, 27]. This is also consistent with what is known of the diversity of fungal associations with terrestrial plants. Members of the Sordariomycetes and Dothideomycetes have been found to be the predominant members of land plant fungal endophyte communities [89], while Eurotiomycetes have been found to be the dominant members of freshwater plant communities [90].

Dark septate endophytes (DSE), particularly members of the Pleosporales within the Dothideomycetes (Fig 6), have been observed to form associations with several seagrass species [33, 42, 44, 71, 72, 91–93]. DSE are a morphological, not phylogenetic (e.g. not each other's closest relatives) group of plant associated fungi, and are largely uncharacterized. The most well described of these DSE associations is between the Mediteranean seagrass, *Posidonia oceanica*, and its dominant root-associated fungus, *Posidoniomyces atricolor*. This Pleosporales member has been found associated with changes in root hair development and can form ecto-

**Table 3. Oomycota isolated from the seagrass, *Zostera marina*.**

| Strain | Isolation Source | Class | Order | Putative Taxonomy | GenBank Accession (SSU) | Top BLAST match | BLAST % Identity | Top BLAST accession no. |
|--------|-----------------|-------|-------|-------------------|------------------------|-----------------|------------------|-------------------------|
| CLE33 | Leaf | Oomycota | Pythiales | *Halophytophthora* sp. | MN944508 | *Halophytophthora polymorphica* | 98.9 | AY598669.1 |
| CLE94 | Leaf | Oomycota | Pythiales | *Halophytophthora* sp. | MN944509 | *Halophytophthora polymorphica* | 98.69 | AY598669.1 |

Here we report the taxonomic information for each oomycete isolate (Class, Order) and the putative taxonomy, provide the GenBank accession number for the 28S rRNA gene sequence for each isolate, and report on the isolation source the isolate was obtained from (e.g. leaf, rhizome, root, seawater or sediment). We also report the taxonomic identity of the top BLAST match against NCBI's nr/nt database, the BLAST % identity to the oomycete isolate and the GenBank accession number for the 28S rRNA gene sequence for the top BLAST match.

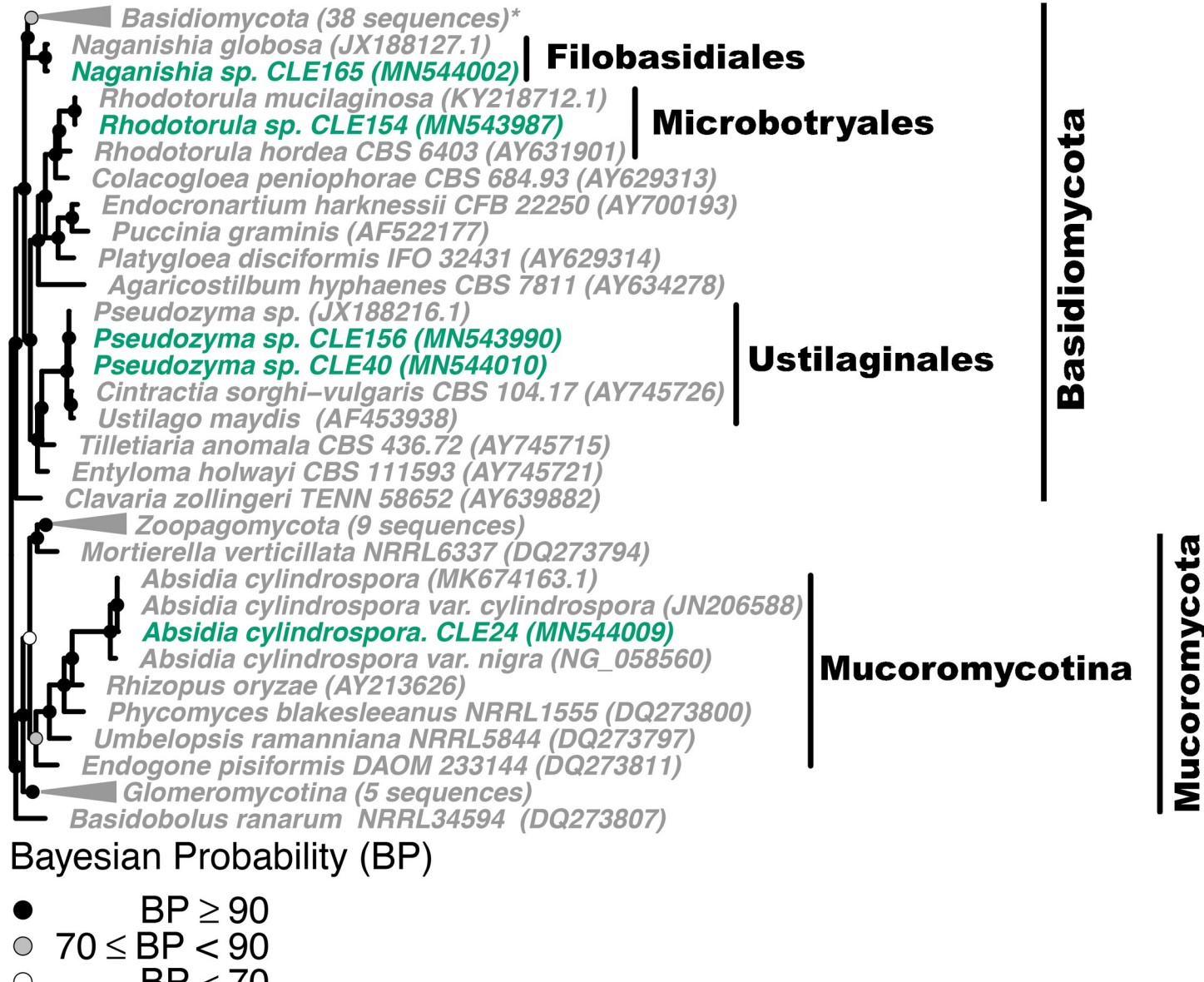

**Fig 3. Phylogenetic placement of seagrass fungal isolates in the Basidiomycota and Mucoromycota.** A molecular phylogeny of 28S rRNA genes for isolates in the Basidiomycota and Mucoromycota was constructed using Bayesian inference. This alignment was generated using MAFFT (v. 7.402) on the CIPRES Science Gateway web server, trimmed using trimAl (v.1.2) and a phylogenetic tree was inferred on the trimmed alignment with a GTR + I + G model using MrBayes (v. 3.2.2) [75–77, 81]. Displayed at each node as a circle in the tree are the Bayesian posterior probabilities (e.g. a black circle represents probabilities greater or equal to 90%, a grey circle represents probabilities greater or equal to 70%, a white circle represents probabilities less than 70%). The names of fungi isolated from *Z. marina* are shown in green, fungi isolated from other seagrass species are shown in black, and all other fungi are shown in grey. For visualization purposes, selected clades have been collapsed and the number of sequences within that clade is indicated. Collapsed clades are shown in green if the majority of sequences in the clade are from isolates associated with *Z. marina*, black if the majority of isolates are from other seagrass species, and grey otherwise. Clade names that are followed by an asterisk contain sequences from multiple sources. An expanded version of this phylogeny can be found in S7 Fig. The GenBank accession numbers of the sequences used to build this phylogeny can be found in Tables 1 and S2–S4.

mycorrhizal-like structures [42, 72, 92, 93]. Here, although we isolated seven members of the Pleosporales, none appeared to be close relatives to *Posidoniomyces atricolor*.

Chytridiomycota were found to be prevalent members of the *Z. marina* leaf microbiome in Ettinger & Eisen [43], however, no chytrids were cultured in this study. This is likely because

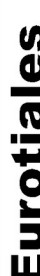

Fig 4. Phylogenetic placement of seagrass fungal isolates in the Eurotiomycetes.

**Bayesian Probability (BP)**

- ● BP ≥ 90
- ◐ 70 ≤ BP < 90
- ○ BP < 70

**Fig 4. Phylogenetic placement of seagrass fungal isolates in the Eurotiomycetes.** A molecular phylogeny of 28S rRNA genes for isolates in the Eurotiomycetes was constructed using Bayesian inference. This alignment was generated using MAFFT (v. 7.402) on the CIPRES Science Gateway web server, trimmed using trimAl (v.1.2) and a phylogenetic tree was inferred on the trimmed alignment with a GTR + I + G model using MrBayes (v. 3.2.2) [75–77, 81]. Displayed at each node as a circle in the tree are the Bayesian posterior probabilities (e.g. a black circle represents probabilities greater or equal to 90%, a grey circle represents probabilities greater or equal to 70%, a white circle represents probabilities less than 70%). The names of fungi isolated from *Z. marina* are shown in green, fungi isolated from other seagrass species are shown in black, and all other fungi are shown in grey. For visualization purposes, selected clades have been collapsed and the number of sequences within that clade is indicated. Collapsed clades are shown in green if the majority of sequences in the clade are from isolates associated with *Z. marina*, black if the majority of isolates are from other seagrass species, and grey otherwise. Clade names that are followed by an asterisk contain sequences from multiple sources. An expanded version of this phylogeny can be found in S8 Fig. The GenBank accession numbers of the sequences used to build this phylogeny can be found in Tables 1 and S2–S4.

the isolation methods used here favor cultivation of Dikarya. Alternative methods and media recipes should be utilized (e.g. baiting) to isolate representatives of these important members of the fungal community. Similarly we note that the methods used here would fail to culture

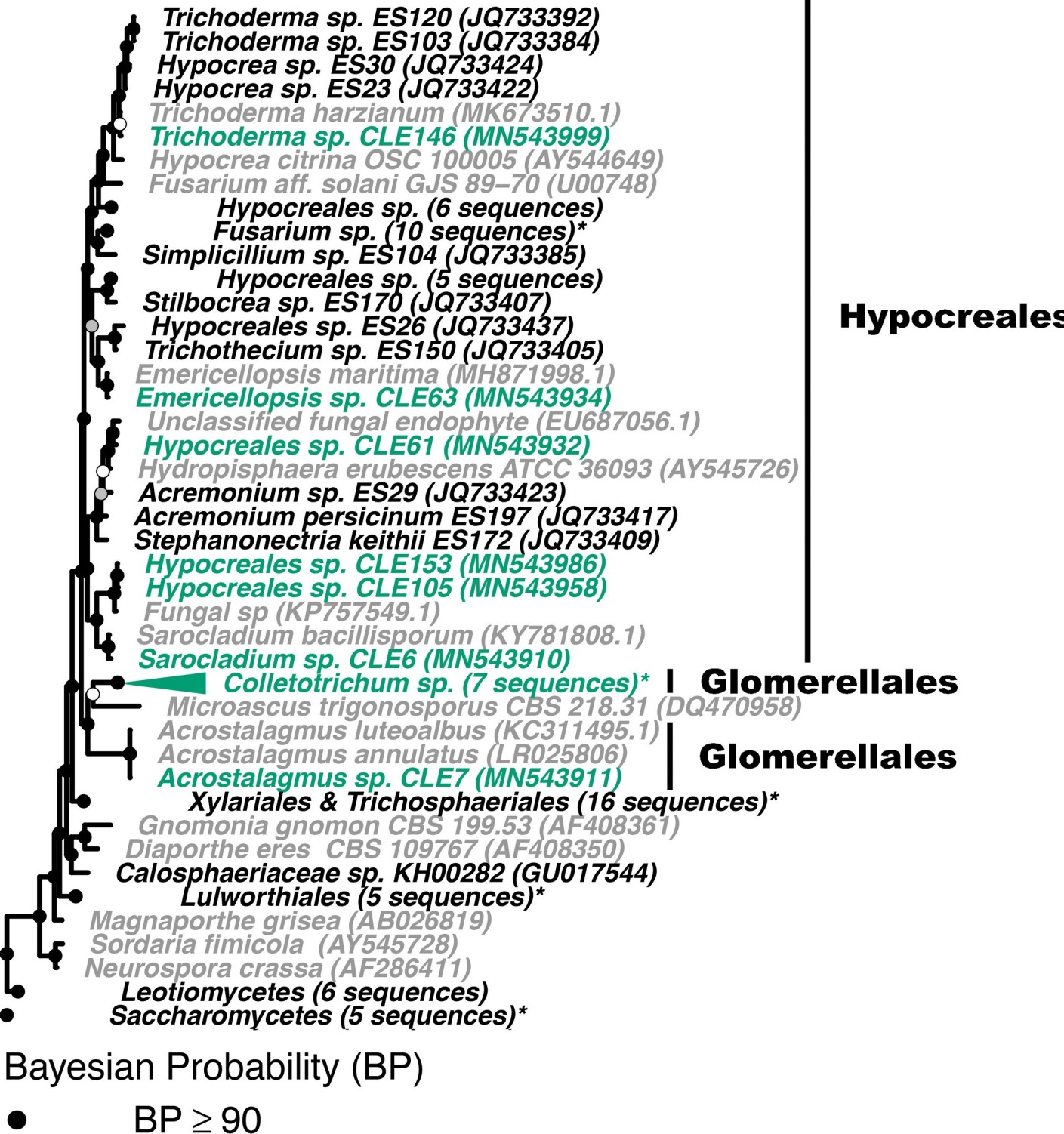

**Fig 5. Phylogenetic placement of seagrass fungal isolates in the Sordariomycetes.** A molecular phylogeny of 28S rRNA genes for isolates in the Sordariomycetes was constructed using Bayesian inference. This alignment was generated using MAFFT (v. 7.402) on the CIPRES Science Gateway web server, trimmed using trimAl (v.1.2)

and a phylogenetic tree was inferred on the trimmed alignment with a GTR + I + G model using MrBayes (v. 3.2.2) [75–77, 81]. Displayed at each node as a circle in the tree are the Bayesian posterior probabilities (e.g. a black circle represents probabilities greater or equal to 90%, a grey circle represents probabilities greater or equal to 70%, a white circle represents probabilities less than 70%). The names of fungi isolated from *Z. marina* are shown in green, fungi isolated from other seagrass species are shown in black, and all other fungi are shown in grey. For visualization purposes, selected clades have been collapsed and the number of sequences within that clade is indicated. Collapsed clades are shown in green if the majority of sequences in the clade are from isolates associated with *Z. marina*, black if the majority of isolates are from other seagrass species, and grey otherwise. Clade names that are followed by an asterisk contain sequences from multiple sources. An expanded version of this phylogeny can be found in S9 Fig. The GenBank accession numbers of the sequences used to build this phylogeny can be found in Tables 1 and S2–S4.

fungi involved in obligate associations with seagrasses. In these cases, a combination of microscopy and/or cell sorting for directed cultivation or sequencing might prove valuable for assessing the functional roles of these organisms to the seagrass ecosystem. Finally, in this study, we only attempted to cultivate aerobic fungi, but there could be anaerobic fungi living in these ecosystems as well.

Previous work on the fungal community associated with the seagrass, *Enhalus acoroides*, identified a pattern of distance decay, where the fungal community was more similar between seagrass that were closer together geographically than between seagrass that were distant from each other [46]. This suggests that dispersal limitation and/or habitat specialization are playing important roles in structuring the fungal community associated with seagrasses. In this study, we opportunistically sampled fungi associated with a single seagrass species, *Z. marina*, from a single seagrass patch in Bodega Bay, CA. We did not investigate the fungal community of this seagrass species at other locations and thus, we cannot test for a pattern of distance decay here. However, we do see some examples of habitat specificity/generalism (and/or dispersal efficiency) at a local level in the fungal genera isolated here from *Z. marina*.

A pattern observed across culture-based studies of seagrass-associated fungi is that ubiquitous fungi are the dominant members of the communities, but that seagrasses also consistently host a diverse set of rare taxa. For example, ubiquitous fungi like *Penicillium* sp. and *Cladosporium* sp. have been previously reported as the dominant fungi of leaves in other culture-based studies of *Z. marina* [36, 38, 94], other seagrass species [26, 34, 35, 41, 95, 96] and freshwater aquatic plants [90]. Additionally, *Penicillium* sp. and *Cladosporium* sp. were some of the only fungi in this study which were found to have close relatives associated with different seagrass species. We hypothesize that these fungal genera may be habitat generalists (taxa that occur evenly distributed across a wide range of habitats [97, 98]) in the seagrass (and potentially larger marine) ecosystem as they were isolated from multiple media types, detected from most sample types (Fig 7) and found to have similar mean relative abundances across sample types (S11 Fig). However, just because these fungi are ubiquitous, does not reflect negatively on their potential importance. These habitat generalists have been shown to be highly adaptable with the innate ability to survive in wide range of extreme conditions (e.g. high salinity), are known to perform ecologically important functions (e.g. degradation of organic matter) and represent sources of biologically interesting and active secondary metabolites [33, 94, 99].

We hypothesize that some fungi associated with *Z. marina* may be habitat specialists (taxa that are more restricted to a specific habitat range [97, 98]). For example, some *Colletotrichum* sp. may be habitat specialists that preferentially associate with *Z. marina* leaf tissue. A *Colletotrichum* sp. ASV (SV10) was found to be dominant on leaves in Ettinger & Eisen [43] and a *Colletotrichum spp.* isolate was previously reported from another seagrass species as a leaf endophyte [95]. However, in this study, we are unable to decouple the contribution of environmental factors (e.g. habitat or niche specialization) from life history strategies (e.g. dispersal, growth rate). For example, although we hypothesize that some genera may be habitat generalists, it is possible that these patterns may also reflect that some genera have more efficient dispersal mechanisms or faster-growth rates and are able to outcompete slower-growing taxa. We realize these ideas

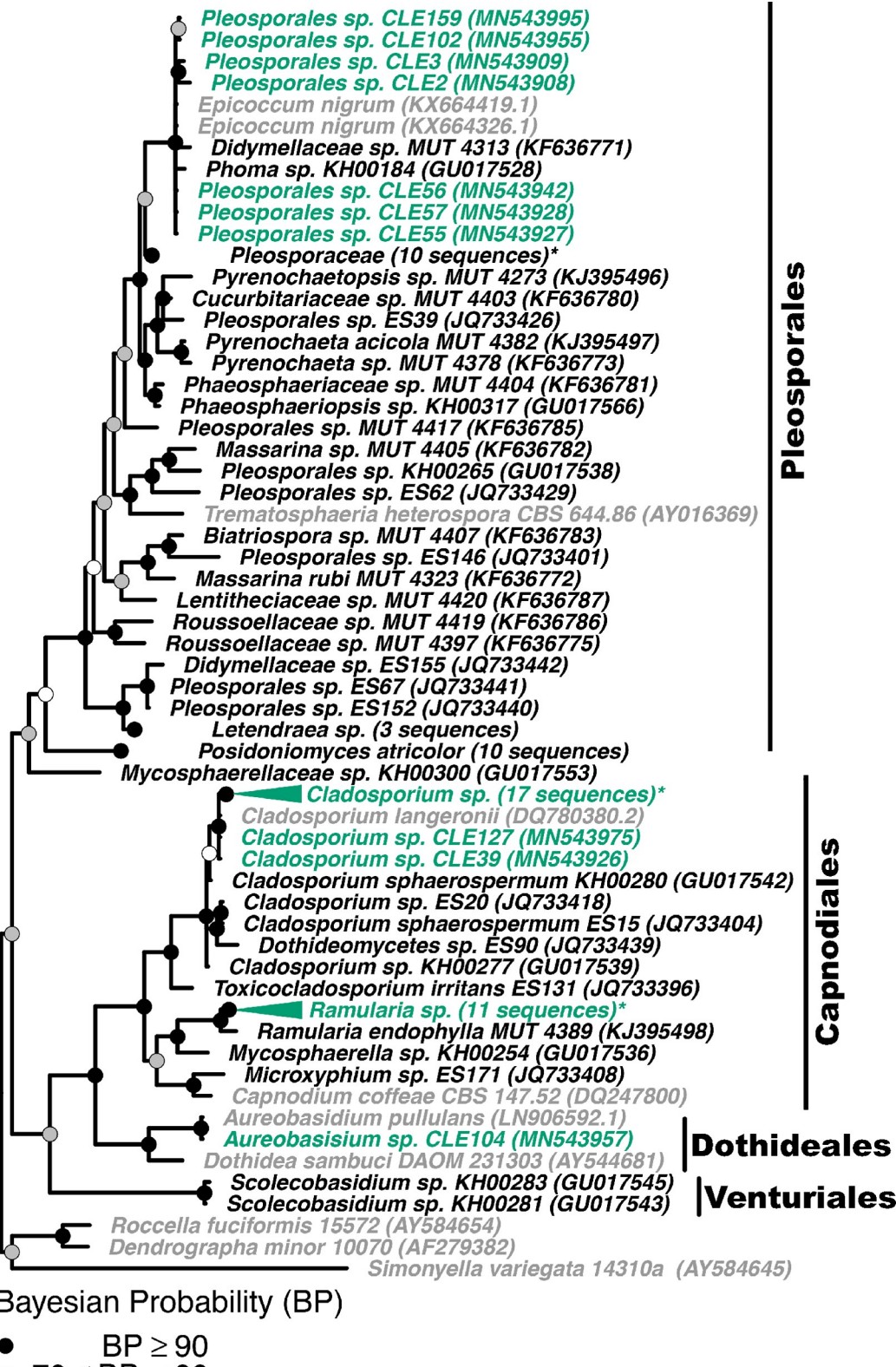

Bayesian Probability (BP)

● BP ≥ 90
◐ 70 ≤ BP < 90
○ BP < 70

**Fig 6. Phylogenetic placement of seagrass fungal isolates in the Dothideomycetes.** A molecular phylogeny of 28S rRNA genes for isolates in the Dothideomycetes was constructed using Bayesian inference. This alignment was generated using MAFFT (v. 7.402) on the CIPRES Science Gateway web server, trimmed using trimAl (v.1.2) and a phylogenetic tree was inferred on the trimmed alignment with a GTR + I + G model using MrBayes (v. 3.2.2) [75–77, 81]. Displayed at each node as a circle in the tree are the Bayesian posterior probabilities (e.g. a black circle represents probabilities greater or equal to 90%, a grey circle represents probabilities greater or equal to 70%, a white circle represents probabilities less than 70%). The names of fungi isolated from *Z. marina* are shown in green, fungi isolated from other seagrass species are shown in black, and all other fungi are shown in grey. For visualization purposes, selected clades have been collapsed and the number of sequences within that clade is indicated. Collapsed clades are shown in green if the majority of sequences in the clade are from isolates associated with *Z. marina*, black if the majority of isolates are from other seagrass species, and grey otherwise. Clade names that are followed by an asterisk contain sequences from multiple sources. An expanded version of this phylogeny can be found in S10 Fig. The GenBank accession numbers of the sequences used to build this phylogeny can be found in Tables 1 and S2–S4.

may not be unconnected and that habitat generalists, by their nature, may be assembled by dispersal related mechanisms, and specialists by species sorting [97]. Regardless, future studies

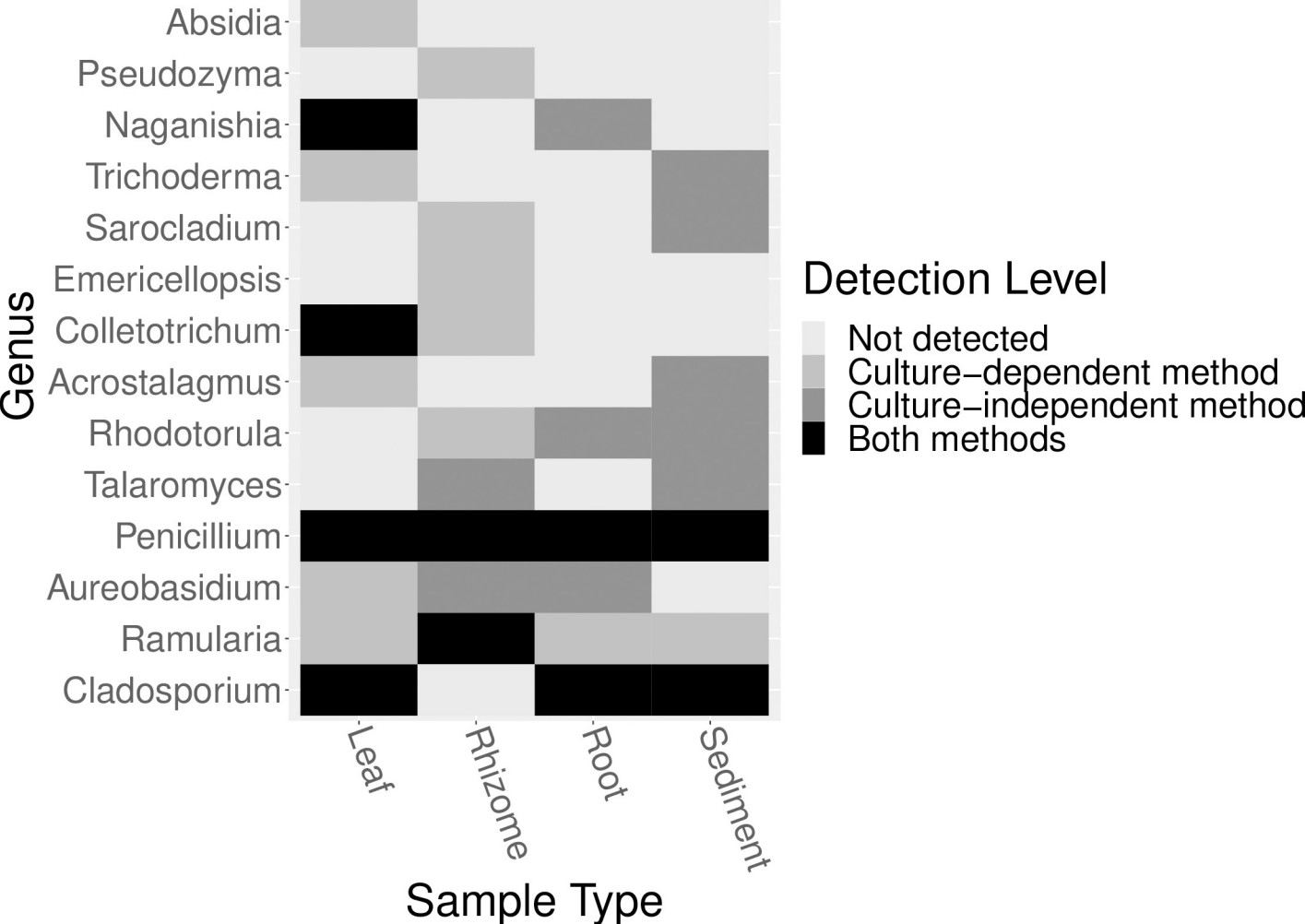

**Fig 7. Comparison of the detection of a fungal genus across methods.** A heatmap representing a comparison of the detection of the presence / absence of fungal genera isolated in this study (using a culture-dependent method) and fungal genera identified in high throughput sequencing data from Ettinger & Eisen [43] (using a culture-independent method). For each fungal genera, we visualize if it was not detected (light grey), detected using only the culture-dependent method (medium grey), detected using only the culture-independent method (dark grey) or detected by both methods (black) for each sample type / isolation source (leaf, root, rhizome, sediment).

could use alternative approaches such as adding different combinations and concentrations of fungicides in order to fully survey rare and slow-growing fungi in this system.

Many of the fungal taxa isolated here are known to have complex life history strategies when associated with land plants. For example, the genus *Ramularia* includes species that are pathogens of a variety of important agricultural plants including barley and sugar beets [100, 101] and the genus *Colletotrichum* includes members that can form endophytic or pathogenic associations with land plants [102]. Additionally there is mounting support for a multi-niche view of fungi, with many phyto-pathogens now being found able to form benign or even beneficial endophytic associations with plants [103]. Thus, future research should endeavour to investigate the true functional roles these fungal genera may have when associated with *Z. marina* and whether these functional roles shift when *Z. marina* is stressed or challenged.

Although our goal here was to isolate seagrass-associated fungi, we also identified 40 bacterial isolates associated with *Z. marina*. Since we were using antibiotics, we do not expect these isolates to be representative of the true diversity of the culturable bacterial community associated with *Z. marina*, just the subset of the community naturally resistant to the antibiotics used here. Most of the bacterial isolates we obtained are from known ubiquitous marine lineages (*Vibrio*, *Pseudoalteromonas*, *Pseudomonas*, *Shewanella*, and *Bacillus*) which are likely habitat generalists and have all been previously cultured from *Z. marina* from Bodega Bay, CA [18, 20, 22–25]. We also isolated several bacterial isolates that may represent rare or slow-growing taxa with interesting ecological implications for the seagrass ecosystem. This includes several Actinomycetes (*Streptomyces* sp., *Rhodococcus* sp. and *Isoptericola)*, representatives of which are known to produce a variety of antibiotics and interesting secondary metabolites [104] and members of the genus, *Isoptericola*, have been previously isolated as endophytes of mangrove plants [105]. We also isolated a *Phyllobacterium* sp. and representatives of this genus are slow-growing $N_2$-fixing plant-growth promoting bacteria which have been previously isolated from mangrove rhizosphere and the roots of land plants [106–108]. Land plants often overcome nitrogen limitation through beneficial relationships with $N_2$-fixing bacteria and similar associations have been observed between $N_2$-fixing bacteria and *Zostera* [14, 109–112]. Based on their role as established $N_2$-fixers in other plant systems, it is possible *Phyllobacterium* sp. may be involved in fixing nitrogen for seagrasses and this possibility should be further investigated.

Just like marine fungi, oomycetes are neglected in marine systems even though they are often implicated as important pathogens of land plants (e.g. *Phytophthora ramorum* [113]), *Phytophthora infestans* [114], *Pythium* spp. [115]). Historically, oomycetes were thought to be fungi based on their similar morphology, but phylogenetic methods have now shown that these organisms are more closely related to diatoms and brown algae [116–119]. In the course of this study, we isolated two members of the *Halophytophthora*. Representatives of *Halophytophthora* have been previously isolated associated with *Z. marina* [47] and this genus includes known saprophytes (organisms living on organic matter) and are thought to be important decomposers in mangrove ecosystems [120]. Recently, Govers et al. [121] suggested that *Halophytophthora* sp. Zostera may be common in *Z. marina* beds, and that this oomycete may serve as an opportunistic pathogen by decreasing seed germination in *Z. marina* populations under certain environmental conditions. More work is needed to understand the possible implications of these oomycetes in the seagrass ecosystem.

## Conclusion

Overall, this study generated a fungal culture collection which broadens understanding of the diversity of *Z. marina* associated fungi and highlights a need for further investigation into the functional and evolutionary roles of fungi and microbial eukaryotes (e.g. oomycetes)

associating with seagrasses more generally. We placed this fungal collection in the phylogenetic context of isolates obtained from other seagrass surveys and found that only habitat generalists were isolated in association with multiple species. We then compared the composition of this fungal collection to high throughput sequencing results of the fungal community associated with *Z. marina* from Ettinger & Eisen [43] and found that taxa isolated here were generally present in the sequencing data, but that they were not prevalent, with the exception of the Glomerellales (e.g. *Colletotrichum* sp.) on the leaves. Although this study adds to general knowledge of the diversity of *Z. marina* associated fungi, there are still many unanswered questions to be addressed related to the life history strategies, functional roles, and dispersal mechanisms of marine and seagrass-associated fungi. One of the biggest challenges in marine mycology is assessing whether the fungal taxa observed are actively growing in the marine ecosystem [122]. For our study here we could ask—are many of the proposed habitat generalists actively growing in the seagrass ecosystem or merely passing through as spores? Additionally, many of the fungi cultured here in association with *Z. marina* have close relatives that are also known to be opportunistic pathogens of land plants. Are these fungi *Z. marina* pathogens or do they serve some other function in the marine environment? Ultimately this work serves as a necessary first step towards experimental and comparative genomic studies investigating the functional roles of these understudied microorganisms and which may lead to important discoveries related to molecular biology, natural product discovery, fungal diversity and evolution, and global importance of marine fungi.

## Supporting information

**S1 Fig. Distribution of counts of inoculated plates across collection trips used for isolation.** A histogram representing the number of inoculated plates faceted by collection trip (October 2017, May 2018, July 2018, August 2018, January 2019), grouped by inoculum source (leaf, root, rhizome, seawater, sediment) and colored by media recipe used for isolation. The media recipes used included 1% tryptone agar, potato dextrose agar (PDA), potato carrot agar (PCA), palm oil media, lecithin media, malt extract agar (MEA), glucose yeast peptone agar (GYPA), and *Zostera marina* agar (Zostera). The numbers included on each bar represent the count of plates inoculated for that media recipe with each inoculum source after each collection trip. For example, the first column shows the count of plates from sampling on October 2017 from leaf samples, with three plates on GPYA and two on ME.
(TIF)

**S2 Fig. Distribution of counts of inoculated plates across tissue treatments and media types used for isolation.** A histogram representing the number of inoculated plates faceted by media recipe, grouped by inoculum source (leaf, root, rhizome) and colored by tissue treatment (tissue wash, crushed tissue, rinsed whole tissue, bleached whole tissue, surface cleaned whole tissue) used for isolation. The media recipes used included 1% tryptone agar, potato dextrose agar (PDA), potato carrot agar (PCA), palm oil media, lecithin media, malt extract agar (MEA), glucose yeast peptone agar (GYPA), and *Zostera marina* agar (Zostera). The numbers included on each bar represent the count of plates inoculated for that media recipe with each inoculum source and tissue treatment combination. For example, the first column shows the count of plates inoculated on 1% tryptone from leaves, with eight plates inoculated with rinsed whole leaves.
(TIF)

**S3 Fig. Distribution of counts of fungal isolates across media recipes used for isolation.** A histogram representing the number of fungal isolates grouped by order and colored by media

recipe used for isolation. The media recipes used included 1% tryptone agar, potato dextrose agar (PDA), potato carrot agar (PCA), palm oil media, lecithin media, malt extract agar (MEA), and glucose yeast peptone agar (GYPA). The numbers included on each bar represent the count of isolates grown on each media recipe.
(TIF)

**S4 Fig. Scatterplots showing observed trend between the count of unique media types, salt sources and isolation sources from which a fungal genus was isolated.** Scatter plots representing A) the relationship between the count of unique isolation sources (leaf, root, rhizome, sediment) and the count of unique media types (PDA, palm oil media, lecithin media, MEA), a fungal genus was isolated from ($R^2$ = 0.86), B) the relationship between the count of unique isolation sources and the count of unique salt sources (no salt, varying amounts of instant ocean [8 g, 16 g, or 32 g], seawater) a fungal genus was isolated from ($R^2$ = 0.92), and C) the relationship between the count of unique media types and the count of unique salt sources a fungal genus was isolated from ($R^2$ = 0.87).
(TIF)

**S5 Fig. Distribution of counts of fungal isolates across tissue treatments used for isolation.** A histogram representing the number of fungal isolates faceted by inoculum source (leaf, root, rhizome), grouped by order and colored by tissue treatment (tissue wash, crushed tissue, rinsed whole tissue) used for isolation. The numbers included on each bar represent the count of isolates in each Order grown with each inoculum source and treatment combination. For example, the first column shows the count of isolates in the Capnodiales from leaf tissue, with six isolates obtained from crushed leaves, one from leaf washes, and four from rinsed whole leaves.
(TIF)

**S6 Fig. Distribution of counts of bacterial isolates across isolation sources and media recipes used for isolation.** Histograms representing the number of bacterial isolates grouped by order and colored by isolation source (A) or media recipe (B). A) When colored by isolation source (leaf, rhizome, root, seawater or sediment), the numbers included on each bar represent the count of isolates obtained from that particular isolation source. B) When colored by media recipe used for isolation (1% tryptone agar, potato dextrose agar [PDA], palm oil media, lecithin media, malt extract agar [MEA], and glucose yeast peptone agar [GYPA]), the numbers included on each bar represent the count of isolates grown on each media recipe.
(TIF)

**S7 Fig. Phylogenetic placement of seagrass fungal isolates in the Basidiomycota and Mucoromycota.** A molecular phylogeny of 28S rRNA genes for isolates in the Basidiomycota and Mucoromycota was constructed using Bayesian inference. This alignment was generated using MAFFT (v. 7.402) on the CIPRES Science Gateway web server, trimmed using trimAl (v.1.2) and a phylogenetic tree was inferred on the trimmed alignment with a GTR + I + G model using MrBayes (v. 3.2.2) [75–77,81]. Displayed at each node as a circle in the tree are the Bayesian posterior probabilities (e.g. a black circle represents probabilities greater or equal to 90%, a grey circle represents probabilities greater or equal to 70%, a white circle represents probabilities less than 70%). The names of fungi isolated from *Z. marina* are shown in green, fungi isolated from other seagrass species are shown in black, and all other fungi are shown in grey. For visualization purposes, selected clades have been collapsed and the number of sequences within that clade is indicated. The GenBank accession numbers of the sequences used to build this phylogeny can be found in Tables 1 and S2–S4.
(TIF)

**S8 Fig. Phylogenetic placement of seagrass fungal isolates in the Eurotiomycetes.** A molecular phylogeny of 28S rRNA genes for isolates in the Eurotiomycetes was constructed using Bayesian inference. This alignment was generated using MAFFT (v. 7.402) on the CIPRES Science Gateway web server, trimmed using trimAl (v.1.2) and a phylogenetic tree was inferred on the trimmed alignment with a GTR + I + G model using MrBayes (v. 3.2.2)[75–77,81]. Displayed at each node as a circle in the tree are the Bayesian posterior probabilities (e.g. a black circle represents probabilities greater or equal to 90%, a grey circle represents probabilities greater or equal to 70%, a white circle represents probabilities less than 70%). The names of fungi isolated from *Z. marina* are shown in green, fungi isolated from other seagrass species are shown in black, and all other fungi are shown in grey. The GenBank accession numbers of the sequences used to build this phylogeny can be found in Tables 1 and S2–S4.
(TIF)

**S9 Fig. Phylogenetic placement of seagrass fungal isolates in the Sordariomycetes.** A molecular phylogeny of 28S rRNA genes for isolates in the Sordariomycetes was constructed using Bayesian inference. This alignment was generated using MAFFT (v. 7.402) on the CIPRES Science Gateway web server, trimmed using trimAl (v.1.2) and a phylogenetic tree was inferred on the trimmed alignment with a GTR + I + G model using MrBayes (v. 3.2.2) [75–77,81]. Displayed at each node as a circle in the tree are the Bayesian posterior probabilities (e.g. a black circle represents probabilities greater or equal to 90%, a grey circle represents probabilities greater or equal to 70%, a white circle represents probabilities less than 70%). The names of fungi isolated from *Z. marina* are shown in green, fungi isolated from other seagrass species are shown in black, and all other fungi are shown in grey. The GenBank accession numbers of the sequences used to build this phylogeny can be found in Tables 1 and S2–S4.
(TIF)

**S10 Fig. Phylogenetic placement of seagrass fungal isolates in the Dothideomycetes.** A molecular phylogeny of 28S rRNA genes for isolates in the Dothideomycetes was constructed using Bayesian inference. This alignment was generated using MAFFT (v. 7.402) on the CIPRES Science Gateway web server, trimmed using trimAl (v.1.2) and a phylogenetic tree was inferred on the trimmed alignment with a GTR + I + G model using MrBayes (v. 3.2.2) [75–77,81]. Displayed at each node as a circle in the tree are the Bayesian posterior probabilities (e.g. a black circle represents probabilities greater or equal to 90%, a grey circle represents probabilities greater or equal to 70%, a white circle represents probabilities less than 70%). The names of fungi isolated from *Z. marina* are shown in green, fungi isolated from other seagrass species are shown in black, and all other fungi are shown in grey. The GenBank accession numbers of the sequences used to build this phylogeny can be found in Tables 1 and S2–S4.
(TIF)

**S11 Fig. Mean relative abundance of fungal orders isolated in this study across sample types in high throughput sequencing data from Ettinger & Eisen [43].** A histogram representing the mean relative abundance of amplicon sequence variants (ASVs) grouped by order and colored by sample type (leaf, rhizome, root, or sediment). The numbers included on each bar represent the mean relative abundance of the order detected on that particular sample type and only mean relative abundances greater than one percent are shown.
(TIF)

**S1 Table. Collection and media specifications for each microbial isolate associated with *Z. marina.*** Here we report the specifics of the culture media used to initially grow each isolate including the media recipe used, the salt source and amount, and the inclusion of dehydrated

crushed seagrass and of various antibiotics. We also report the collection date of the initial inoculum, whether the inoculum if tissue (e.g. roots, rhizome, leaves) came from tissue washes, crushed tissue or whole tissue, whether the plated inoculum was associated with an individual *Z. marina* plant from a core or multiple *Z. marina* plants from a bag, and finally the DNA extraction kit used to extract DNA from each isolate.
(XLSX)

**S2 Table. Fungal sequences used in molecular phylogenies found based on top BLAST matches to *Zostera marina* associated fungal isolates.** Here we report the GenBank accession number and taxonomic information (Class, Order, Molecular ID) for each fungal 28S rRNA gene sequence obtained based on top BLAST matches to fungal isolates in Table 1 and used here to generate Figs 3–6 and S7–10.
(XLSX)

**S3 Table. Sequences from fungi isolated from seagrasses used in molecular phylogenies.** Here we report information about the fungal 28S rRNA gene sequences used here to generate Figs 3–6 and S7–10 which represent fungal strains previously isolated from seagrasses. We note the seagrass species and tissue material (e.g. leaf, root, matte or rhizomes) the fungus was isolated from, as well as the taxonomic information (Class, Order, Molecular ID, Strain), GenBank accession number and the study of origin for each fungal 28S rRNA gene sequence.
(XLSX)

**S4 Table. Non-seagrass associated fungal isolate sequences from the literature used in molecular phylogenies.** Here we report the GenBank accession number and taxonomic information (Phylum, Class, Order, Species, Strain) for each fungal 28S rRNA gene sequence previously used in James et al [73,74] and used here to generate Figs 3–6 and S7–10.
(XLSX)

**S1 File. R Markdown file of all data analysis performed in R.** An R Markdown file of the code used to generate the figures in this manuscript.
(PDF)

## Acknowledgments

We would like to thank the following undergraduate researchers for their contributions to this work including help with sample collection, fungal isolation and DNA extraction: Tess McDaniel, Katelin Jones, Katie Somers and Neil Brahmbhatt. We would like to thank John J. Stachowicz for use of his scientific sampling permit, California Department of Fish and Wildlife Scientific Collecting Permit # SC 4874.

## Author Contributions

**Conceptualization:** Cassandra L. Ettinger.

**Data curation:** Cassandra L. Ettinger.

**Formal analysis:** Cassandra L. Ettinger.

**Funding acquisition:** Cassandra L. Ettinger.

**Investigation:** Cassandra L. Ettinger.

**Methodology:** Cassandra L. Ettinger.

**Project administration:** Cassandra L. Ettinger.

**Resources:** Jonathan A. Eisen.

**Supervision:** Jonathan A. Eisen.

**Visualization:** Cassandra L. Ettinger.

**Writing – original draft:** Cassandra L. Ettinger.

**Writing – review & editing:** Cassandra L. Ettinger, Jonathan A. Eisen.

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
