## [Decision Letter · Decision Letter 0]

27 Apr 2020

PONE-D-20-08398

All small things considered: the diversity of fungi, bacteria and oomycota isolated from the seagrass, Zostera marina

PLOS ONE

Dear Ms Ettinger,

Thank you for submitting your manuscript to PLOS ONE. Your submission was reviewed by three experts in the field. I agree with their recommendations that it has merit but does not fully meet PLOS ONE’s publication criteria as it currently stands. Therefore, we invite you to submit a revised version of the manuscript that addresses the points raised during the review process.

Please modify your title to better reflect the scope of your study. The sampling choices, processing, and data analysis require better rationale in view of serious concerns expressed by the reviewers. Please improve figures and tables for more accurate representation of your findings. 

Please be fully responsive to the reviewers' comments to the extent possible. The revised version of your manuscript will be sent out for another round of peer-reviews.

We would appreciate receiving your revised manuscript by Jun 11 2020 11:59PM. To enhance the reproducibility of your results, we recommend that if applicable you deposit your laboratory protocols in protocols.io, where a protocol can be assigned its own identifier (DOI) such that it can be cited independently in the future. For instructions see: http://journals.plos.org/plosone/s/submission-guidelines#loc-laboratory-protocols

We look forward to receiving your revised manuscript.

Kind regards,

Vishnu Chaturvedi, Ph.D.

Academic Editor

PLOS ONE

Journal Requirements:

"JAE is on the Scientific Advisory Board of Zymo Research, Inc.

CLE declares that the research was conducted in the absence of any commercial or

financial relationships that could be construed as a potential conflict of interest."

3. Please include your tables as part of your main manuscript and remove the individual files. Please note that supplementary tables (should remain/ be uploaded) as separate "supporting information" files

Reviewers' comments:

Reviewer's Responses to Questions

**Comments to the Author**

1. Is the manuscript technically sound, and do the data support the conclusions?

Reviewer #1: Yes

Reviewer #2: No

Reviewer #3: Partly

2. Has the statistical analysis been performed appropriately and rigorously? 

Reviewer #1: Yes

Reviewer #2: No

Reviewer #3: N/A

3. Have the authors made all data underlying the findings in their manuscript fully available?

Reviewer #1: Yes

Reviewer #2: Yes

Reviewer #3: Yes

4. Is the manuscript presented in an intelligible fashion and written in standard English?

Reviewer #1: Yes

Reviewer #2: Yes

Reviewer #3: Yes

5. Review Comments to the Author

Reviewer #1: GENERAL COMMENTS

The authors present a culture-dependent study of fungal communities associated with Zostera marina resulting in a culture collection of fungi, some bacteria and a few oomycetes, which contributes to our still emerging understanding of marine fungi within seagrass ecosystems. Besides phenotypical characterization of obtained colonies, this research used sequencing and phylogenetic approaches to assess the reliability of the identities assigned to the different colonies, both within the context of seagrasses in general as well as in comparison with a previous culture-independent study of Z. marina. This original research included a variety of classic microbiology methods, high-standard analyses and conclusions that are supported by the data. However, the following general concerns should be considered for the review process to continue. Please also find below my specific comments and suggestions.

1. The rationale behind using three different markers (i.e. ITS-28S, 16S and 28S) should be explained. Why did the authors amplified these if they were pursuing a fungal collection?

2. Even though the methodological approaches are clearly explained, there are missing details that should be included in a revised version of the manuscript.

3. Identification approaches get somehow mixed in the Results section due to the use of confusing/broad terminology (e.g. “molecular”), or just lack of specification (e.g. “Bacterial isolates were identified as belonging to…”). It is not clear whether the authors are talking about visual identifications, or those made based on taxonomic or phylogenetic analyses.

4. The discussion about patterns of distance decay is very interesting but its link with this study needs further explanation. The authors should consider expanding on the evidence they provide that supports a role of habitat specificity and/or dispersal efficiency at a local level.

5. In general, table formatting should be improved.

SPECIFIC COMMENTS

ABSTRACT:

Lines 60-62: It might be worth it to mention that Halophytophthora may also be just saprophytes.

Line 62: This study did not generate a culture collection of bacteria or oomycetes, as these microorganisms were not intendedly isolated, and therefore (as mentioned later in lines 502-504) are not representative of the true diversity of the culturable bacterial/oomycetal community associated with Z. marina.

INTRODUCTION:

Line 75: The Introduction section may benefit from some background information about bacteria and oomycetes associated with seagrasses to support the discussion about these microorganisms, provided later provided in the manuscript (lines 501-521 and 523-532).

Lines 90: “Blue carbon” is not an example of the preceding sentence, which indeed defines what blue carbon is. I suggest changing “e.g.” for “i.e.”.

Lines 111-113: Did the authors find any unidentified isolates? Based on this sentence, one would expect to see at least a proportion of fungi without taxonomic assignments. What about contrasting identities (i.e. different across identification methods)?

METHODS:

Line 130: The authors state they collected “bulk leaf tissue”. Does this mean that tissue from all leaves from a single seagrass shoot was collected or there was a preference for a certain leaf rank?

Lines 132-133: Is there a reason to keep samples in dark (and then incubate plates in dark too, lines 164-165)?

Line 135: Was there any attempt to cultivate anaerobic fungi, which may play important roles within the anoxic conditions typical of seagrass sediments? There are modifications of the plating technique that could help reduce Oxygen conditions (e.g. pour plate method).

Lines 175-179: An explanation of the criteria used to choose different extraction kits is missing. Why PowerSoil was not used for this particular subset of isolates?

Lines 213-215: Were also the LSU, WARCUP and UNITE datasets used for oomycetes?

RESULTS:

Line 313: “A total of 108 fungal isolates were obtained” … From how many initial samples?

Lines 315-317: “Combined” samples are not mentioned in the Methods section.

Lines 338-341: Confusing sentence. Suggest re-phrasing.

Lines 345-346: An explanation of why this happens is required, since the use of antibiotics is specified in the Methods. The (close) phylogenetic relationship or other similarities between fungi and oomycetes should also be mentioned.

Line 351: Where these bacterial isolates identified by sequencing, phylogenetic analysis or visual characterization of colonies? This should be clearly stated here and in similar sentences throughout the manuscript (e.g. line 360, etc.).

Lines 391-392: Where these two genera rare members of the microbiome?

DISCUSSION:

Lines 414-418: Missing a period in between these two sentences. Change “,” before “Members of the Sordariomycetes…”.

Lines 432-439: This paragraph might benefit from a brief discussion on anaerobic fungi and the limitations of the study in this regard. Please refer to my comment on line 135 below.

Lines 449-451: Explain further. Which results suggest such a role?

Lines 523-524: There are good examples of plant pathogenic oomycetes in the literature.

Lines 528-521: Please correct the reference at the beginning of the sentence (it is missing the year) and delete the one at the end (not necessary).

FIGURES AND TABLES:

Figure 1: The legend should specify which plates correspond to the different taxonomical groups (i.e. bacteria, fungi or oomycetes).

Reviewer #2: This study aimed at characterizing the cultivable mycobiota of Zostera marina from a sampling site in Bodega Bay, California. The authors also report some bacteria that were obtained in spite of the antibiotics used against their growth and two conspecific oomycetes obtained by chance.

Q1: Does the title of your MS fully reflect the merit of your study? Does an a priori biased bacterial spectrum resistant to antibiotics + two accidentally obtained conspecific oomycetes (typically parasites) warrant especially the first part of the title, which reads “All small things considered”? Not least, there are many other “small things” associated with seagrass surfaces (Bryozoa, Cyanobacteria, Foraminifera, microalgae, etc.), so I would suggest choosing a more conservative/informative title for your manuscript.

The authors “opportunistically” collected “individual” plants of this seagrass during “several sampling trips between summer 2017 and summer 2018”. They also collected “bulk leaf tissue” and seawater.

Q2: It seems like this MS represents random results of some (actually, how many?) random (“opportunistically”) collections during one year rather than a targeted study aimed, e.g., at the seasonal effect on the respective mycobiota or the effect of different plant tissues. What is then the reason of publishing these apparently random data in the current form? If the aim was to provide reference material for further studies, i.e., in this case nrDNA sequences, I suggest their deposition in the INSD would do the same job. If the aim was to establish some reference collection of fungi, then these must be first somehow characterized taxonomically (=at the species level) and deposited in a way that they are available to other researchers – even this, however, does not need to be connected with this type of publication. If the aim was to compare the cultivable community with your previous NGS data – then I suggest the authors use at least the same way of sampling (i.e., “at three timepoints 2 weeks apart spanning July to August”), otherwise any such comparisons make little sense to me.

Q3: Considering the sampling protocol, I am not sure whether the authors were actually able to distinguish “individual” Z. marina plants. If yes: why did you collect “bulk leaf tissue”? How many individual plants were actually collected during each sampling, how many in total? These are important data that may help to understand the diversity and abundance of the mycobiota per one such “individual” plant.

For isolation of the mycobionts, the authors used several cultivation media. For obvious reasons this is a good idea, despite that the available seagrass studies indicate only a little effect of this variable. However, it is not easy to evaluate the different media effect in this study because some important information is missing.

Q4: Did you always use the whole range of these media, with the same number of replicates? How many replicates (per seagrass individual, tissue, sediment, seawater, etc.) did you actually establish per sampling?

The authors did not surface-sterilize the investigated tissues (rinsing them in autoclaved water is definitely not enough to get rid of the plethora of surface-dwelling organisms, including their resting stadia). This approach prevents recognition of organisms living inside these tissues (intra- or intercellularly) from the only randomly associated ones.

Q5: Was there any particular reason for omitting surface sterilization? One can ask a simple question, is the mycobiota detected in your study indeed somehow specific to Z. marina or is it just a random spectrum that could be detected +- on any other submerged surface at your sampling site?

The isolation experiment lasted only 4 weeks which is definitely a very short period as many marine fungi are notoriously slow growing (some need many months to start to emerge from the seagrass tissues). I understand it may be difficult to wait one year for the results but your sampling actually spanned over one year so at least the first isolations could have been kept for a longer time (perhaps at least 2-3 months).

The authors obtained 108 fungal isolates but again, some important info is missing to fully understand the meaning of this number and its context.

Q6: How many replicates did not yield any isolate? How many isolates did you obtain in total, how many of them were non-axenic? How many did you attempt to identify, in how many cases you failed (low quality signal, chimeric sequences – perhaps a sign of non-axenic isolates)? Indeed, it seems that not all isolates were eventually sequenced (L. 171: “from the majority of isolates”) so one could ask why?

One of the ways how to distinguish randomly associated surface dwellers and spores from, e.g., true endophytes would be to compare results from the plated tissue segments and the wash liquid (L. 135-147 in Materials and Methods). The latter is, however, not stated in Results.

Q7: Why there are no results mentioned for the wash liquid? It is hard to believe that the wash liquid would have produced no fungi.

Q8: The Result section includes data for “combined leaf and root tissues” and “combined root and rhizome tissues” – what do these “combined” treatments mean/represent? They are not mentioned in Materials and Methods.

Most of the isolates obtained (n=59) belonged to Penicillium sp., which are ubiquitous fungi occurring practically everywhere especially in the form of spores – this is probably the answer to the question the authors asked themselves (L. 550-551): “For our study here we could ask - are many of the proposed habitat generalists actively growing in the seagrass ecosystem or merely passing through as spores?”. Does it make sense to attribute these specifically to Z. marina?

Q9: What does it mean when you state that you did not “attempt to control for effort or isolation frequency in this study”? Please be specific.

There was a little overlap between the fungi isolated here and in other studies and one of the explanations was that this could be “the result of the limits of culture-dependent studies generally or the use of different media recipes and methods for isolating fungi in other studies.”

Q10: What are the “general limits of culture-dependent studies” that could have caused these differences? Is there any “general limit” in you study that differs from other studies? I cannot find any particular...

Q11: Ad different media recipes, here is what you state in L. 311-313: “…we used a variety of media types including several which had been used previously to isolate fungi from seagrasses (e.g. PDA (Sakayaroj et al., 2010; Vohník et al., 2016; Supaphon et al., 2017), GPYA (Panno et al., 2013), MEA (Torta et al., 2015)).”. Did you detect any similarities at least for these media (PDA, GPYA, MEA)? What other methods of isolation did the other studies use? As far as I know they were actually very similar to your methodology…

The authors wrote that “The fungal taxa that did have close relatives that were associated with other species [in other studies] included Penicillium sp….”/“Penicillium sp. and Cladosporium sp. were some of the only fungi in this study which were found to have close relatives associated with different seagrass species”. Indeed, Penicillium is everywhere and for example in culture-based studies of endophytes, these fast growing sporulating isolates are typically discarded as contaminants. At the same time, the authors note that “Penicillium sp., Cladosporium sp. and Ramularia sp. are drivers of the positive relationship observed previously between the number of tissue types and number of media types a fungal genus was isolated from”…

The main result of this study seems to be the establishment of the collection of the 108 fungal isolates “expanding understanding of the diversity of Z. marina associated fungi”. The collection actually comprises ca. 25 different fungal “species”, ca. 7 of these (more than 70 isolates) are ubiquitous sporulating fungi and 11 are represented by only one isolate – I suggest the authors use a more conservative/modest expression than “to expand”.

Minor comments:

Line 48: endophytes are typically defined as symptomless plant endosymbionts and while I acknowledge that they may have secondary beneficial effects on their hosts, they can be to the same extent harmful, e.g., when environmental conditions change, and typically function along the so-called mutualism-parasitism continuum. An easy way how to solve this might be deleting “beneficial” and perhaps also listing mycorrhizal fungi (which are not endophytes, e.g., see Wilson 1995 Oikos 73:274–276)

L. 54: Penicillium sp., etc. – “sp.” not in italics

L. 58-60: Please refrain from the remarks about “interesting secondary metabolites” and “nitrogen cycling in the seagrass ecosystems” – these claims cannot be supported by any results presented in your manuscript.

L. 81: Comments as for L. 48

L. 135: Please state, at least approximately, the time interval between the sample collection and the tissue plating (the length of this interval may significantly affect the obtained symbiont spectra)

L. 136-7: Please detail the rinsing of tissues in autoclaved water – how many washes, for how much time, etc. In case someone wants to compare your protocol with other studies…

L. 140: Perhaps use 1 ml instead

L. 166-7: This implies that some of the obtained isolates were non-axenic; how did you recognize these and how/on what basis could you be “confident” that the final cultures were axenic?

L. 314: Fig. 1 is nice but redundant and since it is practically the same as Fig. 1 in Amend et al. 2019 I suggest omitting it

L. 359: Z. marina not maraina

L. 426: Mediterranean not mediterranean

L. 427-428: “associated with changes in root hair morphology” – not true, please read carefully the paper by Borovec and Vohnik 2018

References must be doublechecked – for example, there is no journal for Rodriguez 2008 (L. 730), L. 788 Kolařík not Kolarˇík, etc.

Table 1: The column “Genus detected in ITS amplicon data” has a very little information value; for example, Penicillium contains more than 300 species with varying biology so detecting “Penicillium” in both data sets tells nearly nothing…

Figure 4: The 45 Penicillium isolates seem to be conspecific – does it make sense to include all the 45 (practically indentical) sequences in the tree? The same holds true for other non-singleton isolates (perhaps better to establish some representative MOTUs?) and also for reference isolates from other studies (the trees are very large – one representative sequence would be enough).

Reviewer #3: The manuscript by Ettinger and Eisen describes a non-targeted cultivation approach to isolate fungi from Zostera marina. The authors identify the strains using standard techniques and compare the data to their previously published ITS amplicon data. As by-catch, some bacteria and Oomycetes were cultured. The manuscript is straightforward albeit without clear directions or hypotheses that are being tested. Large parts of the data presentation (tables and figures) do not have sufficient quality for publication in their present form.

Comments:

1) Tables 1 and 2 are not in a form that is suitable for publication in a paper. I cant event read some of the column headers. Also, how were these calls made? What were the sequence identity cutoffs, etc. What is a ‘Molecular ID’?

2) Tables 2/3: Top BLAST match. I think the authors can do much better to provide useful information on the ID of the bacteria. E.g., sequence similarity to the most closely related described species, not to a genbank entry with potential dubious history

3) Figure 1: While this looks pretty, it is hardly scientific. The authors should indicate the strain names and why this subset of strains is shown here

4) Figure 4,5,6: Isolates that are identical/nearly identical should get clustered

5) Fig. 7: should show which genera were detected by amplicon sequencing or isolation only. Also, again more detail on how these calls were made is necessary. E.g. are the amplicons 100% matches to the isolates? Same applies to the 'yes' or 'no' for detection of the respective 'genera' in the amplicon data

6) Line 908 and Figure S2: It’s one genus, genera is plural…

7) L. 220: analyzed

8) ‘sp.’ like in Penicillium sp. is not in italics as it’s not a species name. Please correct throughout the paper

6. PLOS authors have the option to publish the peer review history of their article (what does this mean?). If published, this will include your full peer review and any attached files.

Reviewer #1: No

Reviewer #2: No

Reviewer #3: No

---

## [Author Response · Author response to Decision Letter 0]

8 Jun 2020

We have addressed all comments in this revision. Please see the uploaded file titled, "Response to Reviewers", for our point by point comments.

---

## [Decision Letter · Decision Letter 1]

30 Jun 2020

Fungi, bacteria and oomycota opportunistically isolated from the seagrass, Zostera marina

PONE-D-20-08398R1

Dear Dr. Ettinger,

We’re pleased to inform you that your manuscript has been judged scientifically suitable for publication and will be formally accepted for publication once it meets all outstanding technical requirements.

Kind regards,

Vishnu Chaturvedi, Ph.D.

Academic Editor

PLOS ONE

Additional Editor Comments (optional):

Reviewers' comments:

Reviewer's Responses to Questions

**Comments to the Author**

1. If the authors have adequately addressed your comments raised in a previous round of review and you feel that this manuscript is now acceptable for publication, you may indicate that here to bypass the “Comments to the Author” section, enter your conflict of interest statement in the “Confidential to Editor” section, and submit your "Accept" recommendation.

Reviewer #2: All comments have been addressed

Reviewer #3: All comments have been addressed

2. Is the manuscript technically sound, and do the data support the conclusions?

Reviewer #2: (No Response)

Reviewer #3: Yes

3. Has the statistical analysis been performed appropriately and rigorously? 

Reviewer #2: (No Response)

Reviewer #3: Yes

4. Have the authors made all data underlying the findings in their manuscript fully available?

Reviewer #2: (No Response)

Reviewer #3: Yes

5. Is the manuscript presented in an intelligible fashion and written in standard English?

Reviewer #2: (No Response)

Reviewer #3: Yes

6. Review Comments to the Author

Reviewer #2: (No Response)

Reviewer #3: The authors have addressed my concerns appropriately. While the design of the study is not ideal, it does add information to the field.

7. PLOS authors have the option to publish the peer review history of their article (what does this mean?). If published, this will include your full peer review and any attached files.

Reviewer #2: No

Reviewer #3: No

---

## [Editor Report · Acceptance letter]

6 Jul 2020

PONE-D-20-08398R1 

Fungi, bacteria and oomycota opportunistically isolated from the seagrass, Zostera marina 

Dear Dr. Ettinger:

I'm pleased to inform you that your manuscript has been deemed suitable for publication in PLOS ONE. Congratulations! Your manuscript is now with our production department. 

Kind regards, 

on behalf of

Dr. Vishnu Chaturvedi 

Academic Editor

PLOS ONE